# A meteo-hydrological modelling system for the reconstruction of river runoff: the case of the Ofanto river catchment

Giorgia Verri[1], Nadia Pinardi[1,2], David Gochis[3], Joseph Tribbia[3], Antonio Navarra[1], Giovanni Coppini[1], and Tomislava Vukicevic[1] *

[1]Centro Euro-Mediterraneo sui Cambiamenti Climatici, CMCC, Italy
[2]Department of Physics and Astronomy, University of Bologna, Bologna, Italy
[3]National Centre for Atmospheric Research, Boulder, USA
*now at: Interdisciplinary Science and Engineering Division, National Water Center, Tuscaloosa, Alabama

*Correspondence to:* G.Verri (giorgia.verri@cmcc.it)

**Abstract.** A meteo-hydrological modelling system has been designed for the reconstruction of long time series of rainfall and river runoff events. The modelling chain consists of the mesoscale meteorological model Weather Research and Forecasting WRF, the Land Surface Model NOAH-MP and the hydrology-hydraulics model WRF-Hydro. Two 3-months periods are reconstructed for winter 2011 and autumn 2013, containing heavy rainfall and river flooding events. Several sensitivity tests were performed along with an assessment of which tunable parameters, numerical choices and forcing data most impacted on the modelling performance.

The calibration of the experiments highlighted that the infiltration and aquifer coefficients should be considered as seasonally dependent.

The WRF precipitation was validated by a comparison with rain gauges in the Ofanto basin. The WRF model was demonstrated to be sensitive to the initialization time and a spin-up of about 1.5 days was needed before the start of the major rainfall events in order to improve the accuracy of the reconstruction. However, this was not sufficient and an optimal interpolation method was developed to correct the precipitation simulation. It is based on an Objective Analysis and a Least Square melding scheme, collectively named OA+LS. We demonstrated that the OA+LS method is a powerful tool to reduce the precipitation uncertainties and produce a lower error precipitation reconstruction that itself generates a better river discharge time series.The validation of the river streamflow showed promising statistical indices.

The final set-up of our meteo-hydrological modelling system was able to realistically reconstruct the local rainfall and the Ofanto hydrograph.

## 1 Introduction

The problem of reconstructing long time series of river runoff for climate impact studies and management purposes has been dealt with different ways in the literature, from statistical and spatially lumped methods to the most recent deterministic and fully distributed reconstructions (Todini, 1995; Menabde et al., 2001; Todini and Ciarapica, 2002; Rigon et al., 2006; Jones et al., 2008; Ruelland et al., 2008). Here we show a deterministic modelling approach applied to a small river catchment in the

Puglia region (Italy). The scientific community is making considerable efforts to increase the performance of high resolution meteorological and hydrological models as well as to build up coupled modelling systems. However, modelling the spatial and temporal distribution of the water cycle, especially in case of extreme events, is both a research and operational challenge because the water cycle includes several processes which interact between each other and span a wide range of spatial and temporal scales.

Hydrology modelling has gained a limited success in the past due to a lack of input data or model shortcomings. The precipitation records are crucial for assimilation, initialisation and validation procedures, but need to be collected with a fine spacing of stations. A rigorous and detailed representation of soil type and topography conditions is required as they locally modulate the synoptic-scale weather and affect the basin drainage. In addition, numerous and complex air-land and subsurface processes are involved in the local water cycle of river catchments, but cannot be directly measured.

Relatively higher resolution data on topography, land use, soil types and river routing have become available in the last decade. At the same time, the Quantitative Precipitation Forecast (QPF) has been significantly improved (Cuo et al., 2011). Ensemble approaches have been developed to embed the different sources of errors into a probabilistic forecasting (Buizza et al., 2008; Davolio et al., 2013).

Several open issues affect both the meteorological and hydrological modelling. QPF remains one of the most critical tasks for mesoscale meteorological models since the precipitation field is the end result of many multi-scales and interacting processes and is sensitive to topography, soil types and land use conditions (Krzysztofowicz, 1999; Hapuarachchi et al., 2011; Zappa et al., 2011). The grid spacing of mesoscale meteorological models does not allow to fully resolve the scales of the single convective cells/systems (Moeng et al., 2007; Shin et al., 2013). Moreover the quality of meteorological modelling is also critical for ensuring the quality of hydrological modelling as the uncertainties associated with the meteorological unknowns propagate into the hydrological models (Pappenberger et al., 2005; Zappa et al., 2010).

Finally, an additional source of uncertainty is due to the parameterisation of many of the physical processes involved in the water cycle, e.g. water infiltration through the soil column, groundwater drainage and the aquifer water storage (Krzysztofowicz, 2011; Gupta, 2005). These parameterisations imply many tunable coefficients which play an important role in modulating the soil water distribution.

Several hydrological models have been developed in the last few decades: from simple empirical models to shallow water systems with different levels of approximation, i.e. kinematic, diffusive or fully dynamics wave equations.

Two hydrological models that are currently extensively used for operational purposes are: the Hydrologic Engineering Center's Hydrologic modelling System, HEC-HSM (USACE, 2015), developed by the US Army Corps of Engineers and the TOPographic Kinematic APproximation and Integration,TOPKAPI (Todini and Ciarapica, 2002; Liu and Todini, 2002). The latter is currently employed for operational discharge forecasting by the Italian Civil Protection at ARPA-SIM (Agenzia Regionale Prevenzione e Ambiente- Servizio IdroMeteorologico). HEC-HSM and TOPKAPI are both rainfall-runoff models and are based on the kinematic wave approximation of shallow water equations which solve the channel streamflow. No additional representation of 2D overland water flow is considered neither the gravitational drainage nor the coupling with a Land Surface Model, LSM.

Nickovic et al., 2010 propose the HYdrology surface runoff PROgnostic Model (HYPROM), which to our knowledge is the only hydrological model that includes fully prognostic shallow water equations for representing both the 2D overland water flow and the 1D channel streamflow. On the other hand, no representation of the subsurface physical processes is provided by HYPROM.

A more complex hydrology-hydraulics model is the fully distributed WRF-Hydro system (Gochis et al., 2013), originally designed as the hydrological extension of the WRF atmospheric model (Skamarock et al., 2008). WRF-Hydro is based on the diffusive wave approximation for representing both the 2D overland water flow and the 1D river streamflow. In addition, the WRF-Hydro system fully solves the subsurface soil physics and is 2-way coupled with the NOAH-MP land surface model (Niu et al., 2011). Due to all these valuable features we used this model in our study.

The aim of this study is to evaluate the ability of our integrated modelling system to simulate the water distribution in the catchment of the Ofanto river, which flows through southern Italy. Our meteo-hydrological modelling system was set up for the first time in this basin. Two simulations were carried out over the winter 2011 and autumn 2013. Several rainfall events as well as dry periods characterize southern Italy during the selected time ranges. The final goal of this work is a reliable reconstruction of the local rainfall and river runoff thus the evaluation of the modelling performance was carried out focusing on both precipitation and streamflow prediction. We describe both the precipitation reconstruction and the hydrograph results since the former drive most of the quality of the hydrology of a river basin. Critical issues such as the precipitation modelling skill and the calibration of both NOAH-MP and WRF-Hydro models are discussed. A novel approach for the correction of the modelled precipitation is also presented.

The paper is organised as follows. Section 2 describes the study area. The modelling system and the experimental set-up are presented in Section 3. Section 4 discusses the modelling results. Conclusions are drawn in Section 5.

## 2 The study area

The basin of the Ofanto river, flowing through southern Italy and ending in the Adriatic Sea, was chosen as case study in order to test the meteo-hydrological modelling chain we implemented over the Central Mediterranean area (left picture in Figure 1) with a focus on southern Italy (right picture in Figure 1). This is intended to be a relocatable case study as the final configuration of our meteo-hydrological modeling chain may be easily applied to investigate rainfall and runoff events in other study areas with similar physiographic characteristics.

The Ofanto river is a semi-perennial river, whose discharge is close to zero during the dry season, but may significantly increase when heavy rain events occur and eventually cause the river to flood. The mean annual discharge at its outlet is around $15 \ m^3 s^{-1}$, minimum monthly climatology is $2.27 \ m^3 s^{-1}$ in August and reaches its monthly peak, $35 \ m^3 s^{-1}$, in January. The local annual mean rainfall is about 720 mm with large precipitation gradients over small spatial and temporal scales; the annual mean temperature is around $14°C$ (Romano et al., 2009). The watershed area (Fig.2), covering Campania, Basilicata and Puglia in southern Italy, is about 2790 $km^2$ making it a medium-sized catchment (between 1000 $km^2$ and 10000 $km^2$ ), the mean topography slope is $8\%$, and the total length is around 170 $km$ making it the second longest river in southern Italy.

The river source is located south of Torella dei Lombardi, a small village located at 715 m above the sea level. This is not the only source of the river, there are few tributaries with a lower water volume which prevent the bed from becoming dry. The Ofanto basin consists of two distinct areas: the north east and the south west. The southwestern part, representing the upstream reach of the river, is mainly mountain/hilly owing to the Apennine range; the northeastern part, representing the downstream reach of the river, is a flat area which includes the flooding area of the river (see top panel of Fig.2). The predominant soil type in the upstream sub-basin is "loam" according to the United States Geological Survey, USGS, dataset while a less pervious type,"clay loam" category, prevails in the downstream sub-basin (see bottom-middle panel in Fig.2). The dot markers in the bottom panels in Fig.2 indicate four monitoring points along the river network and by using an ArcGIS tool the Ofanto basin can be divided into four sub-basins (bottom-left panel in Fig.2). In particular, the Calitri gauge separates the uppermost sub-basin corresponding to a karst aquifer with respect to the three downstream sub-basins where a porous aquifer is located and favours the salt water intrusion from the Adriatic Sea.

This case study is challenging in terms of both meteorological and hydrological modelling purposes. With regard to the hydrological modelling, this study deals with a medium-size catchment with a "rain-runoff" response time which varies from several days up to a few hours, which makes the streamflow prediction highly demanding. Concerning the meteorological modelling, the case study is located in the Southern Italy, where several heavy rainfall and flash flood events have occurred in the last decades triggered by lee cyclogenesis and convective instability (Federico et al., 2008 and 2009; Moscatello et al., 2008; Miglietta et al., 2008; Mastrangelo et al., 2011).

In September 2000 a severe flood occurred in Soverato, a small town along the Ionian coast of Calabria (Bellecci et al., 2003). Another high impact storm occurred over the Calabria region on 10-12 December 2003 (Federico et al., 2008). An intense weather storm hit southern Italy and in particular the Puglia region on 12-14 November 2004 (Mastrangelo et al., 2011). A flash-flood episode affected a small area in Puglia on 22 October 2005 (Miglietta et al., 2008). A tropical-like cyclone affected south-eastern Italy on 26 September 2006 (Moscatello et al., 2008; Laviola et al., 2011). The small scales of motion involved, meso-$\beta$ and meso-$\gamma$ scales, make it difficult to numerically model these extreme events.

Two recent heavy rainfall events in 2011 and 2013 with consequent flooding of the Ofanto river, are discussed in this paper.

## 3 The Method

### 3.1 Experimental design of the meteo-hydrological modelling system

A meteo-hydrological modelling system was implemented to reconstruct the Ofanto river runoff and is presented here for the first time.

The model chain consists of the meteorological mesoscale model Weather Research and Forecasting WRF v3.6.1, the Land Surface Model NOAH-MP and the hydrological model WRF-Hydro v2. NOAH-MP works as a sub-model of the WRF and WRHydro system and is coupled in a two-way mode with both of them. The WRF and WRF-Hydro system are coupled 1-way. It is worth highlighting that the meteorological and hydrological models coupled 1-way have not been found to produce weaker performances than fully coupled systems. Senatore (2015) recently proved that with the current state of development, the 1-

way or 2-way couplings between WRF and WRF-Hydro show a comparable performance especially in terms of precipitation simulation.

The various modules of the modelling chain and how they face each other are shown in Figure 3.

A detailed description of the equations and parameterisations which are relevant for the discussion of our results is provided in Appendix A.

The WRF model is a widely-used mesoscale meteorological model which solves fully compressible, non-hydrostatic Euler equations. The model adopts a terrain-following hydrostatic pressure vertical coordinate. We chose 58 unevenly spaced levels and set the top of the model at 50hPa. The two domains nested in two-way mode were considered as: a coarse domain covering the Central Mediterranean area with a horizontal resolution of 6km and a inner domain over southern Italy with a 2km horizontal resolution (the domains are depicted in Figure 1).

The two domains set-up (Figure 1) aims to capture the genesis and the development of the mesoscale cyclonic patterns responsible for the heavy rain events in the coarse domain, moreover the finer grid mesh of the inner domain enables to reconstruct the local convection including the orography effects in the region of interest, i.e. the South-Eastern Italy. We tested different extensions and grid spacing of the coarse domain and we compared the 2-domains approach with the 1-domain only set-up. We found that the two-way coupling mode improves the reconstruction of precipitation at local scales (not shown)

The analyses fields built by ECMWF-IFS (European Centre for Medium-Range Weather Forecasts -Integrated Forecasting System) with a 16km horizontal resolution and 6h frequency were adopted as WRF initial and boundary conditions.

A set of sensitivity tests (not shown) highlighted that the terrestrial datasets, i.e. the topography elevation and the land use categories, strongly affect the air-land fluxes and the near surface atmospheric fields. Thus, the default USGS datasets with 800m resolution for topography and land use have been upgraded with higher resolution and more recent data: Corine 250m Land Use categories and EUDEM 30m topography data both released by the European Environmental Agency (EEA); SRTM 90m topography released by NOAA is adopted only over those regions (i.e. northern Africa) not covered by the EUDEM dataset.

In addition, different numerical schemes for the parameterised atmospheric processes have been tested and compared by evaluating how they affect the simulation of the near surface atmospheric fields. The final model configuration uses the RRTMG scheme (Iacono et al., 2008) for long-wave and short-wave radiation, the Monin-Obukhov scheme (Monin et al., 1954) for representing the surface sub-layer of the Planetary Boundary Layer (PBL), and the Yonsei University Scheme (YSU) is the non local K-profile scheme (Hong et al., 2006) that represents the PBL mixed sub-layer. The microphysics was based on the Thompson Double-Moment 6-class scheme (Thompson et al., 2008) for both domains. The cumulus-convection parameterization was based on the Kain-Fritsch scheme (Kain et al., 1993) in the coarse domain while no convection scheme, meaning that the convection is assumed to have been solved explicitly, was found to perform better in the inner domain. Our sensitivity tests shows that the explicit convection works better than the convection parameterization in the inner domain, as its grid spacing is in the "convection permitting" scale range (Prein et al., 2015). This is an expected result, largely documented by previous studies on severe convective weather forecasts: Done et al. (2004), Weisman et al. (2008), Kain et al. (2008), Schwartz et al. (2009) and (2010), among the others.

Table 1 summarizes the terrestrial datasets and parameterisation schemes we adopted as a result of the sensitivity tests.

Overall our experimental design is based on the past studies of WRF for local rainfall events in the same region that stressed the two-way nesting: Miglietta et al. (2008), Moscatello et al. (2008), Federico et al. (2008), Laviola et al. (2011), Mastrangelo et al. (2011) among the others.

The NOAH-MP land surface sub-model was used to solve the partition of the surface water into an infiltration rate and surface runoff together with the water content and temperature of 4 soil layers up to 1m below the ground level. The surface runoff is represented as the excess of surface infiltration capacity, while subsurface runoff is the gravitational drainage at a 1m soil depth, i.e. through the bottom of the solved soil column. The snow modeling is also active in NOAH-MP model: a multilayer snow pack, the snow albedo, the melting/refreezing capability are solved by NOAH-MP. Moreover the evaporation component coming from the snow sublimation is added and the evaporation component coming from the canopy water is split into the rainfall and the snowfall terms. The ECMWF analyses used for computing the initial and boundary conditions provide also the accumulated snow depth at the groundlevel. For our case studies, the snowfall and the melting processes do not seem to play a crucial role.

The added value of the whole WRF-Hydro system with respect to the NOAH-MP is the ability to laterally route both the surface and subsurface water flow, as well as representing their interaction. The surface runoff is routed by a 2D shallow water system (eq. A6-A8) and is also refined on the catchment area: the channel network gains water inflow from both the surface runoff and the aquifer discharge, and the channel streamflow is solved by 1D shallow water system (eq.A11-A12).

We set up the WRF-Hydro model over the WRF inner domain. A detailed catchment routing grid is computed using a GIS procedure starting with EUDEM 30m topography data. The catchment grid is reproduced with a high level of accuracy (bottom-right panel in Figure 2): the drainage directions are first drawn and the river network is then refined by identifying all the branches and the hierarchy of tributaries using Strahler's method (1952). We chose a grid spacing equal to 200m which is ten-times higher than the WRF and NOAH-MP spatial grid. The aquifer water storage is switched on and the aquifer grid is assumed to identically match the watershed grid; 4 sub-basins (bottom-left panel in Figure 2) are defined as the areas located upstream of the monitoring points set along the river network, which enabled us to customise the coefficients of the aquifer recharge/discharge over smaller areas.

Two simulations were performed over January-March 2011 (hereafter "Experiment 1") and November-December 2013 (hereafter "Experiment 2"). The selected time windows included several rainfall events of different intensities. The strongest weather storms occurred on 1 March 2011 and 1 December 2013, hereafter referred to as "Event 1" and "Event 2", and were followed by the flooding of the Ofanto river. Various features of the experiments are summarised in Table 2.

Figure 4 shows the concatenation procedure we adopted for both experiments: a chain of 72h long simulations was carried out and the reinitialization option was chosen for WRF, while a restart option was adopted for WRF-Hydro. We chose a frequent reinitialisation strategy following previous studies carried out with regional climate models on seasonal and sub-seasonal scales (Qian et al., 2003; Koster et al., 2010; Lucas-Picher et al., 2013). These studies highlighted the benefits of working with a concatenation of short simulations rather than a standard continuous simulation. Above all the reinitialisation mitigates the problems of systematic errors and numerical drift and thus improves the accuracy in reproducing the local scale precipitation.

The hydraulics component of WRF-Hydro system is initialized with the NOAH MP overland and subsurface water flows that

are dry at the initial time. Thus a spin-up period is required to laterally route the groundwater of the basin and to allow the river network to reach a steady state. Senatore et al. (2015) considered monthly spin-up for evaluating the WRF-Hydro results and we decided to follow the same strategy.

## 3.2 A two-step correction of the precipitation: objective analysis mapping and observation-model merging

The simulation of the localisation, amount and timing of precipitation is crucial for the reconstruction of a river runoff time series but uncertainties are large in mesoscale models, particularly due to unresolved meso-$\beta$ and meso-$\gamma$ scale processes. In our experiments the horizontal resolution of the WRF inner domain was 2 km. This is quite a high resolution for mesoscale modelling and considers the convection as explicitly resolved by the model. Sensitivity tests done at an early stage confirmed that the "explicit convection" worked better than any convection scheme. However, a few kilometers resolution is a "gray-zone" for representing the convection, because at these scales the power spectrum of the turbulence reaches its peak (Moeng et al., 2007; Shin et al., 2013). This means that the WRF model does not fully reproduce the convective motions and consequently neither reproduces the rainfall events with very local scale features.

In order to increase the performance of the precipitation reconstruction by WRF, we developed a two-step correction algorithm. First we used an Objective Analysis (OA) technique to address the statistical interpolation of the scattered precipitation observations. Mathematical details on the OA technique and the calibration of the OA parameters are provided in Appendix B. Secondly a Least Squares melding algorithm was developed in order to merge the OA optimal estimate with the modelled precipitation. The assimilated precipitation is thus given by the following correction formula:

$$P_a = P_b + \frac{\sigma_b^2/\sigma_f^2}{\sigma_o^2/\sigma_f^2 + \sigma_b^2/\sigma_f^2}(P_o - P_b) \tag{1}$$

where $P_b$ is the modelled precipitation, $P_o$ is the OA optimal estimate, $P_a$ is the corrected precipitation. In addition, $\sigma_o^2/\sigma_f^2$ is the normalised variance associated to the OA (formula B4 in Appendix B) with values in the [0,1] range as shown by the bottom panel of fig.B1, $\sigma_b^2/\sigma_f^2$ is the normalised variance associated with the modelled precipitation also with values in the [0,1] range. Inside the Ofanto basin, the model normalised variance was defined for each grid point as the standard deviation of the modelled precipitation divided by its maximum value, Hereafter, our two-step correction procedure based on OA mapping + Least squares formula will be referred to as the "OA+LS" method.

Overall, we found that the OA+LS method is robust for the correction of the precipitation field. Panels in Fig.5 show the increased performance of the corrected precipitation with respect to the in situ rain gauges data for two days inside Exp1 and Exp2. The left panels in Fig.5 show the simulated precipitation field, the right panels provide the corrected precipitation field. The daily observed precipitation is shown by the stations network. There are 27 rain-gauge stations managed by the Civil Protection of Puglia Region which are regularly distributed over the whole catchment as shown in the top panel of Fig.2. The records cover the whole simulation periods with a 30-minute frequency. It should be noted that a quality control of the observations was performed and two stations (named Cerignola and Borgoliberta') were removed from the validation of Experiment 2 as they differed too much from the surrounding stations.

The added value of the OA+LS method is further discussed and quantified in the next section.

## 4 Results and Discussion

The evaluation of the performance of our modelling system focuses on two fields: the precipitation and the river streamflow. We will show the calibration of the tunable coefficients involved in the parameterisation schemes and discuss the validation of the modelling results.

Critical issues such as uncertainties in the precipitation and the required spin-up of the meteorological simulations are stressed. This study highlights that an integrated modelling system, which includes both the surface and subsurface runoff along with the aquifer water storage, is required to obtain a reliable reconstruction of heavy rainfall and flooding in the Ofanto catchment.

### 4.1 Mesoscale hydro-meteorological features

Figure 6 and Figure 7 provide the mesoscale maps of the two severe weather events occurred on March $1^{st}$ 2011 (Event 1) and December $1^{st}$ 2013 (Event 2).

The 500hPa geopotential maps highlight how the upper level features affect the lower level cyclogenesis. WRF maps for Event 1 show a strong trough of low pressure at 500hPa centred over the Western Mediterranean Sea (left panel in Figure 6) which is due to a cold front (not shown) progressing eastward. At lower levels a strong synoptic wind, coming from the southeast and blowing over the warm Ionian Sea, reaches the Italian Peninsula (right panel in Figure 6), a weak cyclonic pattern is centred over the Tyrrhenian Sea with an associated sea level pressure gradient of 10 hPa moving in the middle of the core (blue patch of the left panel in Figure 6). The left panel of Figure 7 shows the 500hPa geopotential maps for Event 2: a weak trough covers the Western Mediterranean Sea and the Atlas region in the upper troposphere, with a small but deep core south of Sicily. This corresponds to a strong cyclonic circulation at a lower level (right panel Figure 7) with a sea level pressure gradient reaching 16 hPa in the cyclone eye. This cyclone is situated almost directly beneath the cutoff low in the 500hPa height field and corresponds to a southerly winds carrying warm-moist air reaching the Southern Italy and a colder wind developing downslope of the Balkans.

These cyclones triggered Event 1 and Event 2 over the southern Italy with heavy local rainfall and river flooding. As detailed in the Puglia Civil Protection reports, anomalous rainfall hit the Puglia region during both events. A precipitation peak of 186.9 mm/day was recorded on March $1^{st}$ 2011 (Event 1) at the Quasano station in the middle of Puglia, exceeding its historical maximum value of 116mm/day reached in 2010. On December $1^{st}$ 2013 (Event 2) another anomalous precipitation was recorded with 189.6 mm/day (77% fell in only 12 hours) at the Bovino station in northern Puglia compared to a historical maximum value of 135.6mm recorded in 2003. Many other gauge stations reached their absolute maximum rainfall on December $1^{st} - 2^{nd}$ 2013 (e.g. gauges at Quasano, Orsara di Puglia, Cassano Murge, Orto di Zolfo and Castel del Monte).The Ofanto river flooded a few days after both rain events and the recorded water level at the Cafiero gauge station reached 4.62 m on $6^{th}$

March 2011 and 6.48 m on $7^{th}$ December 2013, close to the historical maximum value of 6.8 m recorded on November $11^{th}$ 1929. The Cafiero station was damaged after the peak on $7^{th}$ December 2013 because of the flood, thus the actual peak may have been higher.

## 4.2 The Precipitation field

### 4.2.1 The sensitivity to the initialization time

The simulation of the precipitation field may suffer from a initialisation time that is particularly close or far from the occurrence of precipitation peak events. In the first case the model is unable to develop the mesoscale features required to trigger the local weather pattern, and in the second case the numerical drift may affect the simulation results (Fiori et al, 2014). We avoided the second case by using a concatenation procedure which consists of a chain of WRF 72-hour long and re-initialized simulations. In addition to Experiment 1 and Experiment 2 we performed extra WRF 72h runs focusing on specific events to test the sensitivity of the simulated precipitation in relation to the initialization time: the panels of Figure 8 highlight the differences between the 24h cumulated precipitation on February $18^{th}$ 2011 started 14 hours and 38 hours before the rain peak of Event 1. The left panel shows the 24h cumulate precipitation modelled by WRF with initial time set 14h before the rain peak as it results from the chain of 72-hour long simulations of Experiment 1, the right panel shows the 24h WRF precipitation by choosing 38 hours as the lead time. Overall, we found that the WRF ability to correctly reproduce rainfall events is lowered by a initialization time that is too close to the peak events.

We conclude that our WRF model would need to be re-initialized approximately 1.5 days earlier than the start of the heavy rain events to increase skill in the predicition of precipitation. For this reason as a future step we plan to develop a robust WRF ensemble, which consists of overlapping chains of 72h simulations with a delayed start-time.

### 4.2.2 Validation of the precipitation

For a comprehensive assessment of the QPF performance, we calculated the average BIAS, CORR and coefficient of RMSE variation, CV(RMSE), across all stations. it should be noted that the CV(RMSE) is computed as the root mean square difference between modelled and observed values, i.e. RMSE, divided by the model standard deviation. We believe that the CV(RMSE) indicator gives more rigorous information on the accuracy of the numerical results than the RMSE and the NRMSE (i.e. the RMSE divided by the mean observed value). This is because it weights the model-observation scatter with respect to the variance of the model timeseries. This also makes the comparison between experiments performed over different time ranges with different extreme events more meaningful.

Table 3 summarises the statistical indices: we consider the 24h cumulated precipitation at the model grid points nearest to the basin gauge stations (top panel of Fig.2). Experiment 2 shows a higher BIAS (computed as the modelled minus the observed value) than Experiment 1, but also a better correlation and a lower or equal CV(RMSE). This is expected since Experiment 2 is characterised by a first period (i.e. November 2013) of continuous rainfall and a second almost dry period, while a series of

shorter rain events took place during Experiment 1 making the simulation of single events hard.

Similarly to Yucel's (2014) and Senatore's (2015) studies based on the WRF model, we found that our model set up tends to overestimate the local rainfall (positive BIAS). The correction we propose based on OA+LS method strongly reduces this tendency and generates a weak underestimation. The correction procedure reduces CV(RMSE) by $84\%$ in Experiment 1 and
by $64\%$ in Experiment 2 and increases CORR by $41\%$ in Experiment 1 and by $14\%$ in Experiment 2. Thus, our correction procedure proves to have a great impact on the precipitation modelling performance. The statistical indices we found are even better than those obtained by more sophisticated methods in similar studies with the same model. For example the 3DVAR assimilation scheme by Yucel (2014) reduces the precipitation RMSE by $3.3\%$ with respect to the no correction case, and the correlation coefficient is quite low, 0.364 and 0.360 with and without the correction procedure.

Overall, we conclude that in our experiments modelled and corrected WRF precipitation show a good agreement with the gauge stations in the Ofanto catchment, also in comparison with the results of similar studies, e.g. Fiori 2014; Yucel 2014; Senatore et al., 2015; Givati et al., 2016. The correlation of WRF precipitation reached by Yucel (2014) was 0.364 with a 3DVAR assimilation scheme. A relatively high correlation of WRF precipitation, i.e. 0.92, was found by Givati (2016) by increasing the grid spacing up to 3km. This result refers to only the WRF cumulated precipitation between +6h and +30h with respect to the
start time, and each simulation is 30h long with the reinitialisation option. Our model set-up on the other hand, considers 72 hours as the simulation range plus the reinitialisation option, and the validation is performed over the whole simulation range.

## 4.3    The river runoff

### 4.3.1    Calibration of NOAH-MP and WRF-Hydro tunable parameters

A calibration procedure of the tunable coefficients of both NOAH-MP and WRF-Hydro models was carried out in order to realistically reproduce the Ofanto hydrograph. As a first step we adopted an automated calibration procedure, based on the PEST software (Doherty, 2002). This procedure minimizes an objective function, given by the sum of the mean squared differences between the modelled and observed river streamflow, using the Gauss-Marquardt-Levenberg non-linear least squares method. Several tests were carried out and we identified the most relevant parameters to be calibrated in our specific case study.
The coefficients with a high correlation (i.e. $|corr| > 0.9$) or the ones that preserved almost the original values after the PEST tests have been excluded. Thus we reduced the original set of 25 tunable parameters to 7 that are found to play a key role in the Ofanto basin. They are: the surface roughness scaling factor which controls the hydrograph shape and the timing of the peaks; the infiltration coefficient, the saturated hydraulic conductivity and the aquifer coefficients which control the total water volume.

As a second step, we carried out a manual calibration. Although this is a fairly rough approach, it avoids the uncertainties arising from the tuning of highly correlated parameters when a non-linear least squares method such as PEST is adopted.
The optimal values of the WRF-Hydro tuned parameters are listed in Table 4 and Table 5.
Among the WRF-Hydro parameters controlling the hydrograph shape, Manning's 2D and 1D roughness coefficients play a

crucial role as they are involved in the empirical formula used to compute the discharges of both the 2D overland water flow (eq.A9-A10) and the 1D channel streamflow (eq.A14). Manning's 2D roughness coefficients are indexed using the Land Use Categories, while 1D roughness coefficients are assigned on the basis of Strahler's stream order. In order to upgrade the computation of the 2D roughness coefficients, we replaced the default USGS Land Use Categories with the higher resolution and updated Corine data (EEA dataset). We refined also the computation of Strahler's order and thus the 1D roughness coefficients by adopting the higher resolution and updated EUDEM topography data (EEA dataset).

The 2D roughness coefficients can also be calibrated using a scaling factor whose values vary between 0 and 1.0, where values equal to 1.0 mean that Manning's 2D roughness coefficients are not changed. We found this factor needs to be adjusted and we progressively reduced it in order to obtain faster water streaming: 0.05 is the optimal value to best capture the timing of the peaks compared with the observed hydrograph.

The NOAH-MP sensitive coefficients are the infiltration and the saturated hydraulic conductivity. They affect both the surface water budget and the moisture content of the NOAH-MP soil layers through the parameterisation of the infiltration capacity (eq.A4). They also indirectly condition the WRF-Hydro overland water flow through the source term of the shallow water system (eq. A2) as well as the aquifer discharge (eq.A15) through the gravitational drainage of the deepest soil layer (eq.A5). Additional sensitivity tests pointed out the seasonality of the soil physics. We found that the coefficients for the infiltration and the saturated hydraulic conductivity are seasonally dependent and thus different values are assumed in the two experiments. In winter, the soil is expected to be wetter than in autumn, and the soil porosity lower, which implies that the infiltration coefficient value is fixed as lower and saturated hydraulic conductivity as higher in Experiment 1 than in Experiment 2.

The calibration of WRF-Hydro aquifer was also performed manually. The aquifer discharge directly feeds the river streamflow by affecting the river baseflow. Thus, the manual calibration of the aquifer coefficients was carried out by comparing the simulated and observed hydrograph. The tuned values of the aquifer coefficients are listed in Table 5.

Our sensitivity experiments highlighted that the aquifer discharge is soil-type and seasonal dependent. The "Clay Loam" soil type of the Ofanto low valley is much less pervious than the upstream "Loam" soil type. Thus, the low valley (sub-basins 2, 3 and 4 of our catchment) is characterised by a lower hydraulic conductivity which tends to counteract the upward aquifer discharge. We set the values of the initial water depth of the aquifer, the exponential law coefficient and the volume capacity lower in the downstream sub-basins 2, 3 and 4 than in the upstream sub-basin 1, while the maximum depth of the aquifer is higher.

Experiment 1 also shows higher volume capacity coefficient and lower maximum depth than Experiment 2, as the soil column is expected to be wetter and thus the elevation of the water table is higher in winter than in autumn.

This study found that the soil infiltration and the aquifer water storage parametrizations should be seasonally dependent. This means that the present parameterizations of these processes are not capable to capture the complexity of the groundwater physical processes.

### 4.3.2 Validation of the runoff

The observed water level at the Cafiero station was used to validate the river runoff modelled by WRF-Hydro.

To compare the model versus observed river level, we used the "stage-discharge" relationship for the Ofanto river at the Cafiero Station, which converts the model runoff into the water level. This is because the water level as computed by WRF-Hydro is conditioned by the channel geometrical parameters such as side slope, bottom width and channel roughnesses. These parameters are prescribed as a function of the Strahler's stream order instead of being customized for the specific catchment, which makes the model water level unreliable.

Figure 9 shows the observed and modelled hydrograph of the Ofanto river in Experiment 1 by using the simulated precipitation (top panel) or the corrected precipitation (bottom panel). Similarly Figure 10 refers to Experiment 2. The gap in the observed time series in Fig.10 is due to the river flood on December $7^{th}$, which damaged the gauge station.

In both experiments, working with the model precipitation, the river water level tends to be overestimated (top panels of Fig. 9 and 10). This is reduced by the corrected precipitation (eq.1) and thus the runoff peaks are better captured (bottom panels of Fig. 9 and 10).

For an overall assessment of the streamflow reconstruction, we calculated the CV(RMSE) and CORR at the Cafiero station. Table 6 summarises the statistical indices, considering the model grid point nearest to the Cafiero gauge station and a hourly frequency. The first significant result is that the OA+LS method is a powerful way of dealing with the precipitation uncertainties, and it has a positive impact on the hydrological reconstructions. The OA+LS procedure reduces the WRF-Hydro CV(RMSE) by 20% in Experiment 1 and by 6% in Experiment 2 and increases CORR by 24% in Experiment 1 and by 19% in Experiment 2. This is significant also in comparison with similar studies based on WRF-Hydro. For example, Yucel (2015) shows that the assimilation of the precipitation by a 3DVAR method reduces the hydrograph RMSE by 7.6% but does not affect the correlation, which is equal to 0.90 and 0.89 without and with the assimilation of the precipitation field, respectively.

It is important to stress that by validating the Ofanto hydrograph we were able to calibrate the OA tunable coefficients: the optimal values are those that ensure that the simulated hydrograph is the closest to the observed one.

Some shortcomings are evident in the representation of the Ofanto hydrograph. The first problem is the timing of the runoff peak after a severe rain event. If the observed water level peak occurs at times close (less than 24 hr) to the rain peak event, the simulated water level has a low skill. A short lead time (less than 24 hours) is the reason for the delay in the simulated peak centred on November $23^{th}$ 2013 and for the peak underestimation starting on December $2^{nd}$ 2013 (Fig.10), both of which are associated with local-scale rainfall. On the other hand, the peaks observed on February $19^{th}$ 2011 and on March $6^{th}$ 2011 are well captured despite the rain short lead time because they are linked to weather events with large spatial scales.

It should be also noted that the Event 2 onset overlaps the start time of WRF 72h simulation (Table 2) and this probably affects the underestimation of the runoff peak starting on December $2^{nd}$ 2013.

For a comprehensive analysis of the shortcomings in the representation of the Ofanto hydrograph, the limited coverage of the rain gauge stations and the quality of the observed values should be considered. The overestimation of the hydrograph we found in November 2013 (Fig.10), when WRF-Hydro is forced by the corrected precipitation, enabled us to identify and remove some

"outliers" among the observed precipitation values on November $5^{th}$ and $11^{th}$. On the other hand, the overestimation of the hydrograph persists owing to the WRF overestimate of precipitation on November $4^{th}$ and $10^{th}$. The daily maps of modelled and assimilated precipitation in Fig.11 show that the WRF overestimate cannot be removed by the OA+LS corrections as there are no rain gauges in the uppermost region of the Ofanto basin where the rainfall peaks occurred.

Despite the predictability gaps in the precipitation field and the limited coverage of the rain gauges, the final configuration of our meteo-hydrological modelling chain with an appropriate calibration was found to reasonably reconstruct the Ofanto hydrograph and to correctly reproduce the water level peaks as well as the plateaus.

To conclude we compare the water level reconstruction at Cafiero station with and without the acquifer and the same but without the Hydro component of the modelling system. The blue timeseries in Figure 9 shows the water level when the aquifer in

not activated with respect to the red timeseries with the aquifer switched on. The comparison shows that the river baseflow is affected by the acquifer parametrizations and that a better representation is obtained with the aquifer. On the other hand, the acquifer parametrizations do not impact the quality of the reconstruction because of the small Ofanto catchment acquifer capacity, as shown in Fig.9 (i.e. CV(RMSE) index reduces of only 2% when the aquifer is switched on and the correlation is almost the same).

Figure 12 shows that coupling the land surface model NOAH-MP with the WRF-Hydro model component is crucial. The "column only" land surface model NOAH-MP parameterises the surface runoff through eq.(A2) which is inadequate to represent the Ofanto hydrograph as shown by the blue time-series in Fig.12.

## 5   Summary, conclusions and future plans

This study investigated the ability of a new meteo-hydrological modelling system, based on WRF and WRF-Hydro models, to reconstruct the local water cycle of a small catchment in southern Italy.

We chose a challenging case study: the semi-perennial Ofanto river with a small-size catchment and a porous aquifer in the downstream region. The river basin is also located in the southern Italy, which is frequently subject to flash flood events.

We chose two time windows characterized by the occurrence of severe weather events with flooding of the Ofanto.

Our study provides the first implementation of the WRF-Hydro system in this region.

The aim was to highlight the strengths/weaknesses of the final set up of our meteo-hydrological modelling system, as well as to develop a useful tool for reconstructing the water runoff in rivers for several months.

One of the novelties of this study lies in our OA+LS method which we used to correct the modelled precipitation. The Objective Analysis, OA, statistically interpolates the scattered observations on the WRF model regular grid and the Least

Squares method, LS, is used to merge the OA optimal estimates with the modelled precipitation. This is a powerful method to deal with precipitation uncertainties, providing a positive impact on the hydrological reconstructions. The OA+LS procedure improved the precipitation estimate by reducing RMSE by 84% and increasing correlation by 41%, water level connected to runoff was improved with a RMSE reduction of 20% and correlation increase of 24%.

The quality of the modelling system was proven by the validation of both precipitation and water level predictions. We obtained promising statistical indices for both fields, also in comparison with recent studies dealing with WRF and WRF-Hydro models (e.g. Fiori et al., 2014; Yucel et al., 2014; Yucel et al., 2015; Senatore et al., 2015; Givati et al., 2016).

The calibration of NOAH-MP and WRF-Hydro tunable coefficients highlighted that the infiltration and the aquifer coefficients are seasonally dependent. This means we need to account for increased complexity in the parameterisation of the groundwater physical processes.

The performance of our modelling system was affected by various error sources: (i) the predictability limit of the precipitation field using a mesoscale meteorological model due to the meso-$\beta$ and meso-$\gamma$ scales involved in the rainfall events; (ii) the sensitivity of the precipitation predictions in relation to the initialisation time which cannot be too close or far from the rainfall events. The WRF model set-up thus needs a spin-up period of about 1.5 days before the start of the rain event in order to be able to realistically reproduce the local weather pattern, and 72 hours was found to be an adequate reinitialisation range.

Our next step is to exploit the OA+LS method as a flexible technique to correct other WRF variables. This technique could be embedded into an operational meteo-hydrological forecasting system. We are also planning to develop a more robust WRF ensemble which consists of overlapping chains of 72-hour simulations with a delayed start-time.

Overall, we highlighted the 2-way feedback existing between a proper reconstruction of the meteorological events and the hydrological ones. A reliable description of the river hydrograph goes through a proper description of the meteorological and soil processes, with the precipitation field playing the most relevant role. At the same time the validation of the river hydrograph works as a effective post-processing tool to calibrate the water infiltration through the soil column and the aquifer recharge/discharge as well as to correct the modeled precipitation with the OA+LS method.

More research is required to establish a better groundwater modeling that at the moment considers seasonally dependent, ad hoc values of the soil infiltration and the aquifer water storage. We plan to evaluate different parameterizations of the aquifer recharge/discharge. Overall a reduction of the parameterizations involved in the WRF-Hydro system could be desirable.

*Acknowledgements.* This work was funded by the Italian Project TESSA through the Centro Euro-Mediterraneo sui Cambiamenti Climatici, Lecce, Italy. We are very grateful to Dr. Franco Intini from the Civil Protection of Puglia region (http://www.protezionecivile.puglia.it/centro-funzionale) and to Dr. Lia Romano from the Autorita' di Bacino of Puglia region for providing ground-based observations of meteo-hydrological variables. The first author would like to thank Wei Yu and Kevin Sampson of NCAR RAL Division for their valuable technical support.

## Appendix A: NOAH-MP and WRF-Hydro physics

The NOAH-MP land surface model (Niu et al., 2011) is a "column-only" model which solves the vertical routing of surface and subsurface water flow based on 4 soil layers up to 1m below the ground level (layer thicknesses are 0-10cm, 10-30cm, 30-60cm, 60-100cm). A multilayer snowpack is also modelled. Basic equations are the prognostic equations for both the soil moisture content (Richards' equation) and the temperature of the 4 soil layers plus a diagnostic equation for the soil surface water budget. A set of parameterisation schemes are also used to compute the surface energy flux components (Niu et al, 2011), the gravitational drainage at the bottom of the deepest soil layer and the partitioning of the soil surface water (sum of rainfall, dewfall and snowmelt reduced by the evaporation rate) into an infiltration rate and surface runoff (Niu et al., 2007). The parameterisations of the infiltration rate and groundwater drainage are key issues as they are linked to the hydrological modelling performed by WRF-Hydro and are affected by the calibration of soil texture and moisture coefficients. The infiltration rate, I (units of $ms^{-1}$), is computed as:

$$I = min(\dot{H}_{sfc}, F_{frz}I_{MAX}) \tag{A1}$$

and the surface runoff (units of $ms^{-1}$) is parameterised as follows:

$$R = max(0, \dot{H}_{sfc} - F_{frz}I_{MAX}) \tag{A2}$$

where $\dot{H}_{sfc}$ is the current surface water rate (units of $ms^{-1}$) as computed by the surface water budget equation. $F_{frz}$ is the fractional impermeable area as a function of soil ice content of the surface layer, $I_{MAX}$ is the maximum soil infiltration capacity (units of $ms^{-1}$) dependent on soil texture and moisture. The empirical formula is given below:

$$I_{MAX} = H_{max}\frac{C_{inf}}{H_{max} + C_{inf}}/\Delta t \tag{A3}$$

where the maximum surface water level (units of m) is given by $H_{max} = max(0, \dot{H}_{sfc}\Delta t)$ and the infiltration capacity, $C_{inf}$ (units of $m$), at the upper soil layer (k=1) is an empirical function of 4 tunable coefficients (units of $m^3/m^3$): the maximum surface moisture content $SMC_{MAX}$, the minimum surface moisture content required by the plant not to wilt or below which transpiration ceases $SMC_{WLT}$, the surface infiltration coefficient $REFKDT$ and the saturation of soil hydraulic conductivity $REFDK$. The empirical formula is below:

$$C_{inf}(k=1) = [\sum_{k=1}^{N}\Delta z(k=1)(SMC_{MAX}-SMC_{WLT})(1.0-\frac{(SMC(k=1)-SMC_{WLT})}{(SMC_{MAX}-SMC_{WLT})})](1-e^{-\frac{SMC_{MAX}\cdot REFKDT}{REFDK}\Delta t_1}) \tag{A4}$$

where $\Delta z(k=1)$ is the thickness of the upper soil layer, $\Delta t_1 = \Delta t/86400$ is the model time step converted to the ratio of a day, $SMC(k=1)$ is the soil moisture content (units of $m^3/m^3$) of the upper soil layer, as solved by Richards' equation.

The computation of groundwater drainage is also a crucial step since this is assumed to be the recharge flow which feeds the unconfined aquifer below the soil column. The groundwater drainage is assumed to be a free gravitational drainage, $Q_{bot}$ (units of $mms^{-1}$), thus formulated as a function of the current soil moisture content in the deepest soil layer:

$$Q_{bot} = SLOPE \cdot DKSAT \cdot [max(0.01, SMC(k=4)/SMC_{MAX})]^{2 \cdot B + 3} \cdot (1 - F_{frz}) \tag{A5}$$

5    where $SMC(k=4)$ is the soil moisture content of the deepest soil layer, as given by Richards' equation. $DKSAT$ and $B$ are soil type dependent coefficients: the first is the saturated soil hydraulic conductivity (units of $mms^{-1}$), the second is a non-dimensional value accounting for soil texture. Finally $SLOPE$ is a coefficient between 0.1-1.0 which modifies the gravitational free drainage out of the bottom layer depending on the surface slope categories of the grid cells: 9 slope classes are prescribed with a different range of surface percentage slope following Zobler's method (1986).

The WRF-Hydro model (Gochis et al., 2013) was designed as an extension of the "column only" NOAH-MP model with several physics modules which describe the lateral routing of surface and subsurface water flows and how they interact with each other.

The WRF-Hydro system includes 4 routing modules which represent the saturated subsurface flow, the 2D overland water flow,

the aquifer recharge/discharge and the 1D channel streamflow (Figure 3).

The added value of this model is represented by two shallow water (SW) systems which describe both the 2D overland water flow and the 1D channel streamflow. The overland water flow occurs when the surface water level of specific grid cells exceed a fixed retention depth which is assumed to depend on the surface slope. The SW system for the overland water flow is based on the diffusive wave hypothesis, meaning that the inertia term of the momentum equation is neglected. The shear stresses in

the momentum equation are negligible. Overall, the shallow water equations read:

$$\frac{\partial h}{\partial x} - S_{fx} + S_{ox} = 0 \tag{A6}$$

$$\frac{\partial h}{\partial y} - S_{fy} + S_{oy} = 0 \tag{A7}$$

$$\frac{\partial h}{\partial t} + \frac{\partial q_x}{\partial x} + \frac{\partial q_y}{\partial y} = i_e \tag{A8}$$

where the unknowns are the water column thickness $h = h(x,y,t)$ (units of m) defined as the free surface water level minus the bottom topography (i.e. the height of the river bed) $h = \widetilde{h} - h_{bot}$, and the unit discharges (units of $m^2 s^{-1}$) in the x- and y- directions, i.e. $q_x = h(x,y,t)u(x,y,t)$ and $q_y = h(x,y,t)v(x,y,t)$. The sink/source term of the continuity equation, $i_e$, is the surface runoff parameterised by NOAH-MP (as detailed in eq.(A2)). Moreover $S_{fx} = \frac{\nu}{g} \frac{\partial^2 u}{\partial x^2}$ and $S_{fy} = \frac{\nu}{g} \frac{\partial^2 v}{\partial y^2}$ are the non-

dimensional friction slope terms (where $\nu$ is the kinematic viscosity coefficient with units of $L^2 T^{-1}$) and $S_{ox} = \frac{\partial h_{bot}}{\partial x}$ and $S_{oy} = \frac{\partial h_{bot}}{\partial y}$ are the non-dimensional terrain slope terms. Finally, $\frac{\partial h}{\partial x}$ and $\frac{\partial h}{\partial y}$ are the non-dimensional pressure slope terms.

The $S_{fx}$ and $S_{fy}$ terms are computed by analytically solving the momentum equation, where $h$ is assumed to be the overland water level provided by NOAH-MP equations for the surface water budget.

Manning's formula provides the unit discharges $q_x$ and $q_y$ as an empirical function of the water column $h = h(x,y,t)$:

$$q_x = \frac{\sqrt{|S_{fx}|}h^{5/3}sign(S_{fx})}{n} \tag{A9}$$

$$q_y = \frac{\sqrt{|S_{fy}|}h^{5/3}sign(S_{fy})}{n} \tag{A10}$$

where the surface roughness coefficient, $n(x,y)$ (units of $sm^{-1/3}$), is a tunable parameter defined as a function of the land use categories. The unit discharges $q_x$ and $q_y$ are then replaced in the continuity equation and $h = h(x,y,t)$ is numerically solved with the Courant constraint ensuring the stability of the numerical solution.

The diffusive wave equations allow for backwater effects and waterflow on adverse slopes, which represents an added value with respect to the widely-used kinematic wave models which neglect the pressure slope terms.

     The channel streamflow is computed on a pixel-by-pixel basis along the river network grid. The channel network has a trapezoidal geometry, its parameters (side slope, bottom width and roughness coefficients) are "a priori" defined as a function of Strahler's stream order. The river streamflow is activated if river network points intercept the 2D overland waterflow. The gov-

erning equations are based on the same assumptions of 2D overland waterflow including the diffusive wave hypothesis and are written as follows:

$$\frac{\partial A}{\partial t} + \frac{\partial Q}{\partial x} = q_{lat} \tag{A11}$$

$$\frac{\partial h}{\partial x} = -S_0 + S_f \tag{A12}$$

where the unknowns are the volume flow rate $Q = Q(x,t)$ and the wetted area $A = A(x,t)$. The channel water level $z(x,t)$ is derived from $A$ by considering the trapezoidal shape of the channel cross-section: $A(x,t) = (L_{bot} + \alpha z(x,t))z(x,t)$ where $L_{bot}$ and $\alpha$ are the bottom width and the side slope of the channel cross section. Similarly to the 2D shallow water equations, $S_f$ is the friction slope term, $S_0$ is the terrain slope term and $\frac{\partial h}{\partial x}$ is the pressure slope term with $h$ assumed as the water level solved by the 2D continuity equation (A8). Finally $q_{lat}$ is the lateral flow (units of $m^2 s^{-1}$) in (positive) or out (negative) of the

channel and is supplied by the surrounding overland water flow and the aquifer discharge as follows:

$$q_{lat}(x,y,t) = \sqrt{q_x(x,y,t)^2 + q_y(x,y,t)^2} + \frac{Q_{out}}{S_{catch}}h(x,y,t) \tag{A13}$$

with $q_x$ and $q_y$ are computed by the 2D momentum equation (A6-A7) and by taking into account the only overland computational grid points bordering the river points. In the second term on the RHS of (A13), $S_{catch}$ is the catchment area and $Q_{out}$ the aquifer discharge computed by a conceptual unconfined aquifer is located below the bottom layer of NOAH-MP with a

horizontal extension matching the catchment area. The solving strategy is the same as that adopted for the 2D shallow water (eq.A6-A8) with eq.A12 analytically solved to get $S_f$ which is then replaced in Manning's formula for the 1D channel to derive $Q$ as the empirical function of $A$. The Manning formula for the 1D channel is:

$$Q = \frac{A^{5/3}\sqrt{|S_f|}sign(S_f)}{P^{2/3}n} \tag{A14}$$

where $P$ is the wetted perimeter computed as a function of $h = h(x,y,t)$, $n$ is the tuneable coefficient for the channel roughness defined for each branch as a function of Strahler's stream order. The discharge $Q$ is then replaced in eq.A11 which is numerically solved and provides the wetted area $A$.

A sub-model describes the aquifer recharge/discharge. It is forced in 1-way mode by NOAH-MP groundwater drainage, $Q_{bot}$ (eq. A5), and provides the aquifer discharge, $Q_{out}$, by means of the following empirical function:

$$Q_{out} = min(C(e^{\frac{\alpha \cdot z}{z_{max}}} - 1), z \cdot S_{catch}/dt) \tag{A15}$$

where $z$ is the current conceptual water depth of the aquifer given by the sum of the groundwater drainage and the stored aquifer water $z = z + Q_{bot}dt$. Tunable parameters are the initial value of the aquifer water depth $z_{ini}$ (units of $mm$), the maximum value of the aquifer water depth $z_{max}$ (units of $mm$), the exponential law coefficient $\alpha$, and the volume capacity of the aquifer $C$ (units of $m^3/s$).

**Appendix B: Objective Analysis of the precipitation field**

The Objective Analysis (OA) is a statistical estimation of a specific field by interpolating irregularly spaced data over a regular grid on the basis of the Gauss-Markov theorem. It was introduced in meteorology by Gandin (1963) and in oceanography by Bretherton et al. (1976). We adopted the Harvard OA code (Carter and Robinson, 1987).

On the basis of the Gauss-Markov theory, the OA optimal estimate of a target field over a regular grid has the following form:

$$\theta_{\overrightarrow{x}} = \widetilde{\theta_{\overrightarrow{x}}} + \sum_r C(\overrightarrow{x}, \overrightarrow{x_r}) \sum_s [C(\overrightarrow{x_r}, \overrightarrow{x_s}) + IE]^{-1}(\phi_s - \widetilde{\theta_{\overrightarrow{x}}}) \tag{B1}$$

where $\overrightarrow{x_r}$ are the original locations of the irregularly spaced observations $\phi_r$ with $r = 1, ..N$, $\phi_s$ stands for the array of observations interpolated over the regular $\overrightarrow{x}$ grid at locations $\overrightarrow{x_s}$ with $s = 1, ..N$, $< \epsilon_r, \epsilon_s > = IE$ is the normalised variance of the observation errors, $C(\overrightarrow{x_s}, \overrightarrow{x_r})$ is the observation correlation function, $C(\overrightarrow{x}, \overrightarrow{x_r})$ is the model correlation function. Knowledge of the correlation functions is key to the method. The practical procedure by Bretherton et al. (1976) is considered, thus

the correlation functions are expressed by a two-parameter function with one-degree of freedom $C(\overrightarrow{x_r}, \overrightarrow{x_s}) = F(\overrightarrow{x_r} - \overrightarrow{x_s})$ and $C(\overrightarrow{x}, \overrightarrow{x_r}) = F(\overrightarrow{x} - \overrightarrow{x_r})$.

The function $F$ is commonly written as:

$$F(r) = (1 - \frac{r^2}{a^2})e^{-(\frac{r^2}{2b^2})} \tag{B2}$$

where $r$ is the distance grid point-gauge station and is limited by the radius of influence of each station, $R_{inf}$. The parameter $a$ is the decorrelation length (units of km) and $b$ is the decay length (units of km) with the assumption $a \geq (b, R_{inf})$. A constraint of the correlation function in (B2) is the hypotheses of quasi spatial homogeneity and isotropy. We are aware that this assumption is basically inadequate (Lynch et al., 2001) for a critical field, as the precipitation field which is strongly affected by the topography at very local scales. This issue will be investigated further in the future.

$\widetilde{\theta_{\vec{x}}}$ is the OA first guess and is estimated as the weighted average of the observations as follows:

$$\widetilde{\theta_{\vec{x}}} = \frac{1}{\sum_{r,s}[C(\vec{x_r}, \vec{x_s}) + IE]^{-1}} \sum_{r,s}[C(\vec{x_r}, \vec{x_s}) + IE]^{-1}\phi_s \tag{B3}$$

The normalised error variance associated with the OA optimal estimate is given by:

$$(\theta_{\vec{x}} - \widetilde{\theta_{\vec{x}}})^2 = C(\vec{x}, \vec{x}) - \sum_{r,s} C(\vec{x}, \vec{x_r})[C(\vec{x_r}, \vec{x_s}) + IE]^{-1}C(\vec{x_s}, \vec{x}) + \frac{(1 - \sum_{r,s} C(\vec{x}, \vec{x_s})[C(\vec{x_r}, \vec{x_s}) + IE]^{-1})^2}{\sum_{r,s}[C(\vec{x_r}, \vec{x_s}) + IE]^{-1}} \tag{B4}$$

We assume the normalised variance of the observations, E, equal to 0.1 and perform a series of sensitivity tests to set the parameters $a$, $b$ and $R_{inf}$. The final choice is a=20km, b=15km, $R_{inf} = 20km$. The OA mapping was carried out on an hourly basis, i.e. the model output frequency. The panels in figure B1 provide an overview of the method by focusing on the 24h cumulated precipitation on 2011/03/01.

It is worth to note that the validation of local precipitation and Ofanto hydrograph was adopted to calibrate the OA tunable parameters: their final values ensure that the assimilated precipitation and thus the simulated hydrograph are the closest to the available observations.

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

**Figure 1.** The study area. Left panel: WRF coarse domain (EEA-SRTM topography dataset). Right panel: WRF inner and WRF-Hydro domain (EEA-Eudem topography dataset)

| WRF ARW SET-UP | DOMAIN 1 | DOMAIN 2 |
|---|---|---|
| Topography | SRTM 90m + EUDEM 30m (Europe) | EUDEM 30m |
| Land Use categories | USGS 800m + Corine 250m (Europe) | Corine 250m |
| Radiation | RRTMG (2008) | RRTMG (2008) |
| PBL surface sub-layer | Monin-Obukhov (1954) | Monin-Obukhov (1954) |
| PBL mixed sub-layer | YSU (2006) | YSU (2006) |
| Convection | Kain-Fritsch (1993) | Explicit |
| Microphysics | Thompson (2008) | Thompson (2008) |

**Table 1.** Terrestrial datasets and parameterization settings adopted over WRF coarse Domain 1 (6 km grid spacing) and inner Domain 2 (2 km grid spacing)

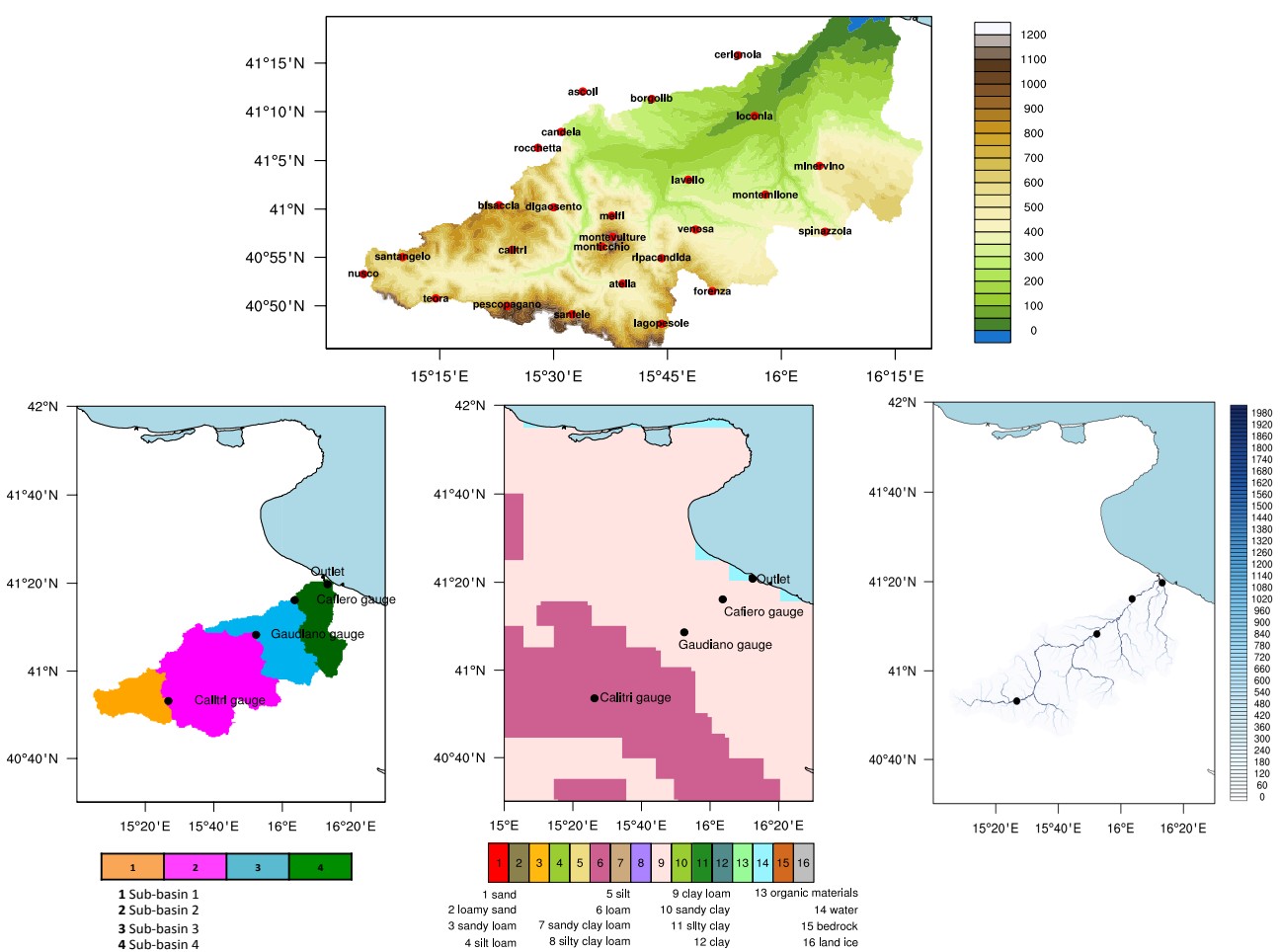

**Figure 2.** The Ofanto river Catchment. Top panel: Topography height (units of m) and location of 27 rain-gauge stations. Bottom left panel: The whole basin and the 4 sub-basins (coloured zones) defined as the areas upstream of the selected monitoring points (black dots). Bottom middle panel: USGS Soil Type Categories in the region of the Ofanto basin. Bottom right panel: The flow accumulation grid defined by the number of grid cells which drain into an individual cell along the river network grid.

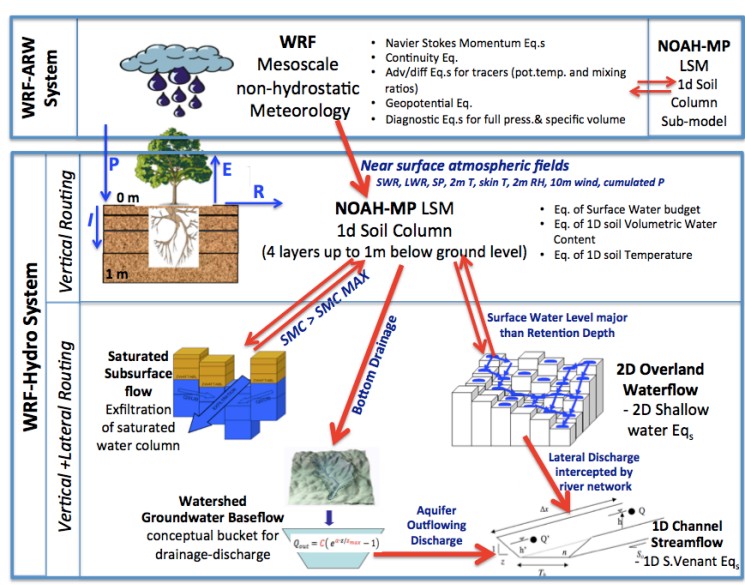

**Figure 3.** The meteo-hydrological modelling chain

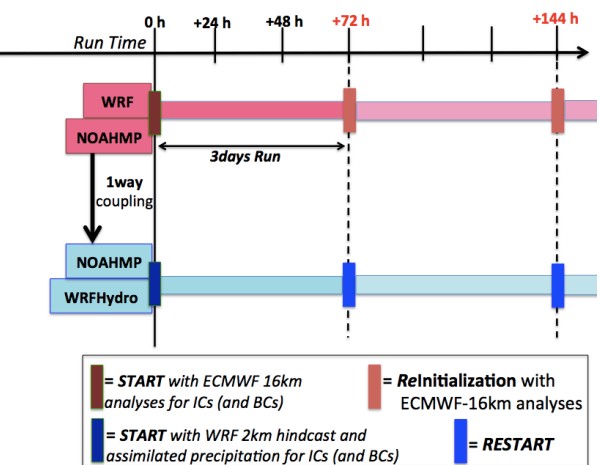

**Figure 4.** The concatenation procedure of the simulations

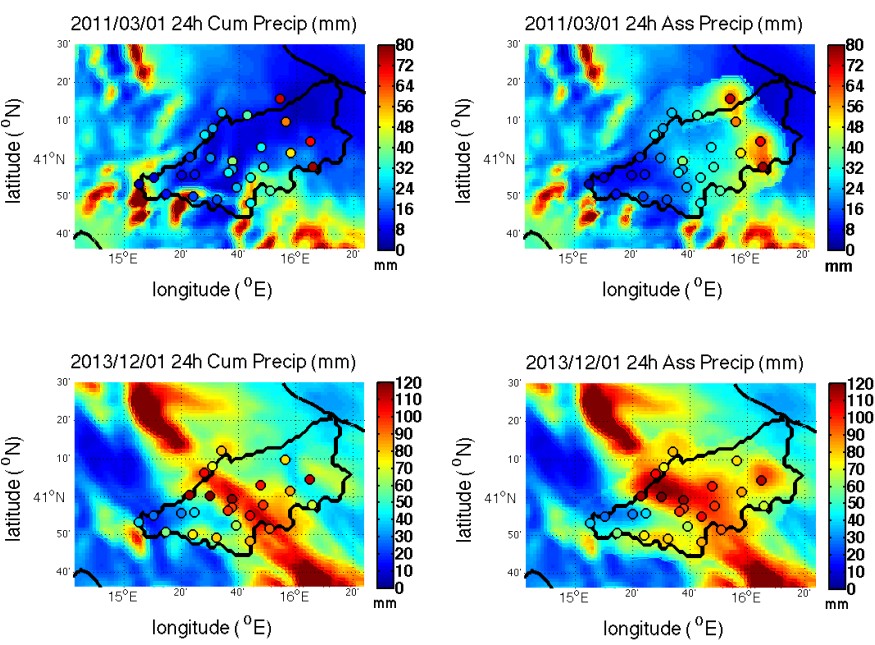

**Figure 5.** Maps of 24h cumulated precipitations (in mm/day, colours) during the peak events on March, $1^{st}$ 2011 (Top panels) and December, $1^{st}$ 2013 (Bottom panels): shaded maps of modelled (left panel), and assimilated (right panel) precipitation with overlapped observed spots over the Ofanto basin

| Experiment | Time Window | Date of Severe Rainfall Events | Start time of WRF 72h Run for Events 1-2 | Maximum recorded value of 24h cumulated precipitation |
|---|---|---|---|---|
| Experiment 1 | Jan-Mar 2011 | 1 March 2011 (Event 1) | 27 February 2011 00 UTC | 186.9 mm/day |
| Experiment 2 | Nov-Dec 2013 | 1 December 2013 (Event 2) | 1 December 2013 00 UTC | 189.6 mm/day |

**Table 2.** Details on the Experiments

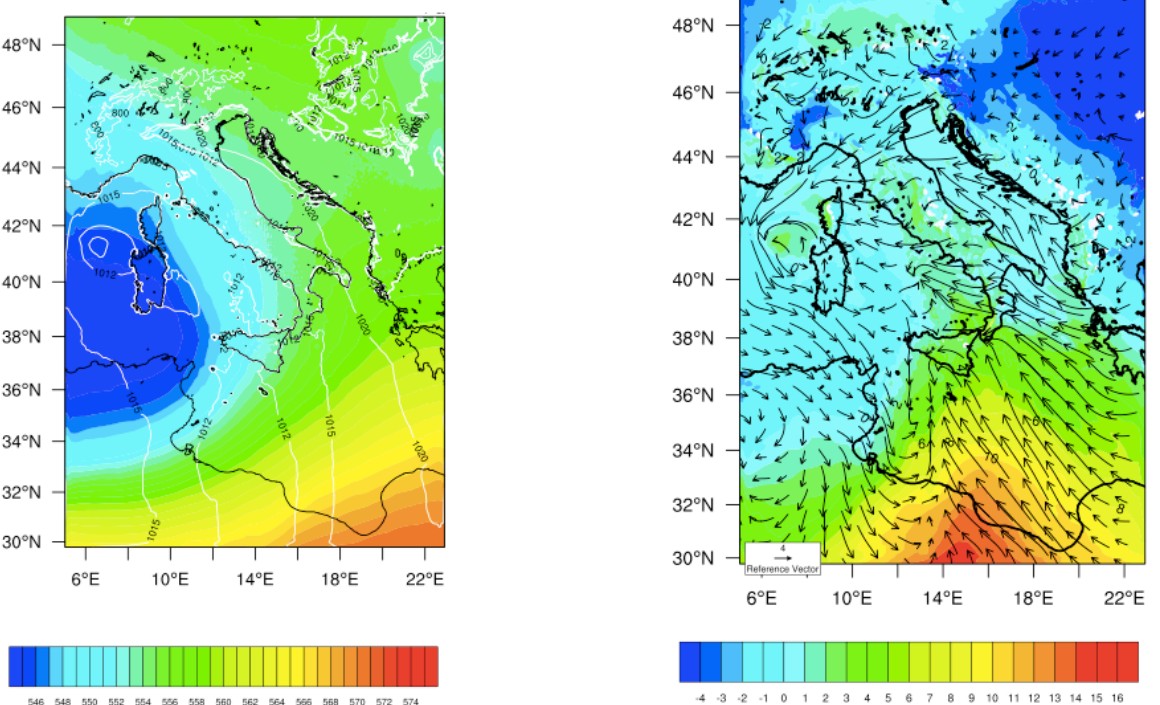

**Figure 6.** Mesoscale maps during the weather storm on 1 March 2011 (Event 1). Left panel: WRF (domain1) Geopotential height (in dam,colours) at 500hPa and mean sea level pressure (in hPa, white lines). Right panel: WRF (domain1) 850hPa Temperature (in C, colors) and 10m wind (in m/s, black arrows)

| Statistical index | modelled Precipitation | | Assimilated Precipitation | |
|---|---|---|---|---|
| of precipitation | Experiment 1 | Experiment 2 | Experiment 1 | Experiment 2 |
| $CV(RMSE)_{ave}$ | 1.15 | 0.56 | 0.18 | 0.20 |
| $BIAS_{ave}$ (mm/day) | +0.11 | +0.48 | -0.06 | -0.20 |
| $CORR_{ave}$ | 0.70 | 0.86 | 0.99 | 0.98 |

**Table 3.** Statistical indices for validation of modelled and assimilated precipitation by comparison with rain-gauge stations in the Ofanto basin

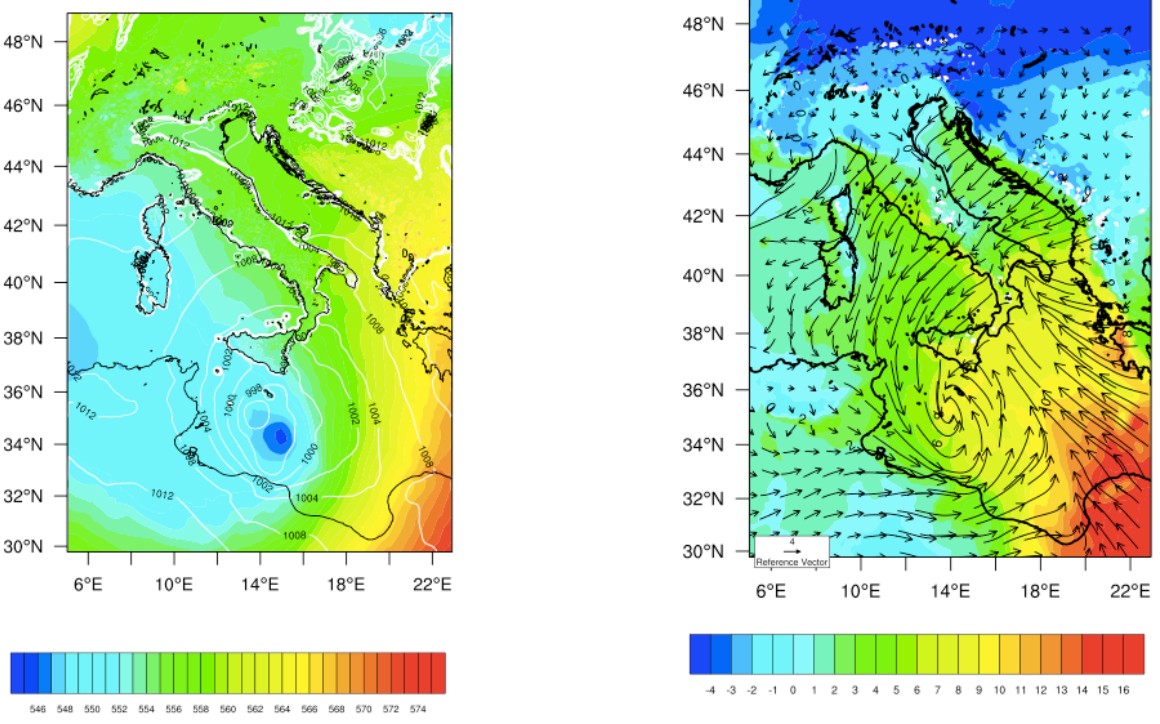

**Figure 7.** Mesoscale maps during the weather storm on 1 December 2013 (Event 2). Left panel: WRF (domain1) Geopotential height (in dam,colours) at 500hPa and mean sea level pressure (in hPa, white lines). Right panel: WRF (domain1) 850hPa Temperature (in C, colors) and 10m wind (in m/s, black arrows)

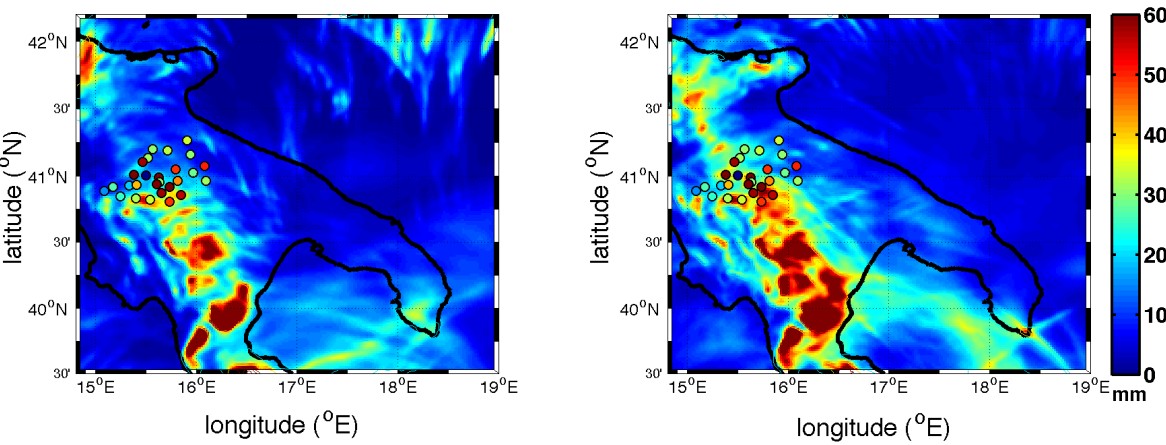

**Figure 8.** Comparison of 24h cumulated precipitation on 2011/02/18 as modelled by WRF with start time 14h before the rain peak (left panel) and 38h before the rain peak (right panel). Recorded values by 27 gauge-stations are overlapped

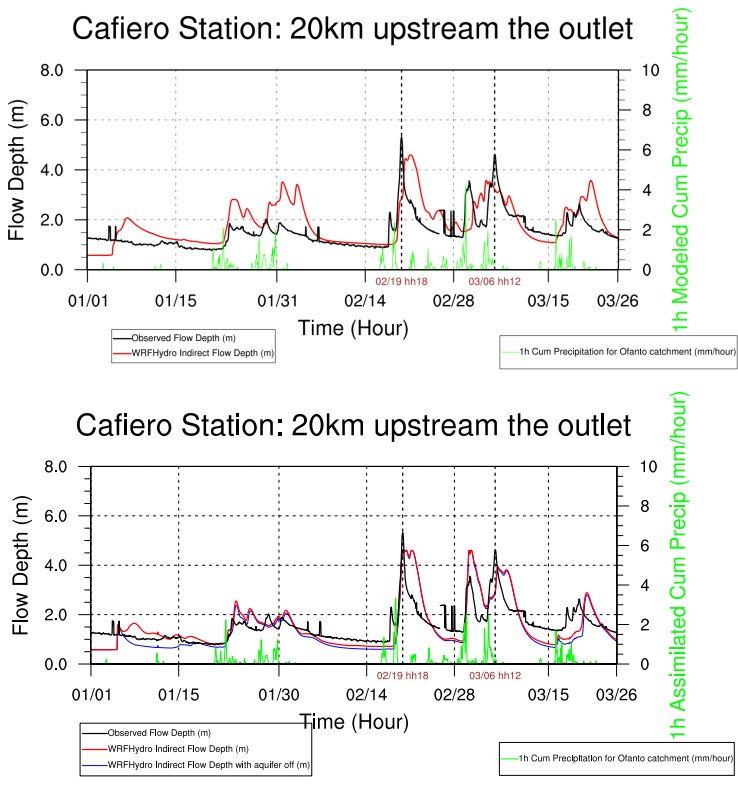

**Figure 9.** Validation of the Ofanto discharge for Experiment 1 at Cafiero Station. Top panel: modelled precipitation. Bottom panel: assimilated precipitation. The blue timeseries refers to the additional experiment performed with the aquifer switched off

| Tuned Coefficient | Experiment 1 | Experiment 2 |
|---|---|---|
| OVROUGHRTFAC | 0.05 | 0.05 |
| REFKDT $(m^3/m^3)$ | 0.2 | 0.6 |
| REFDK $(m^3/m^3)$ | $2.2 * 10^{-6}$ | $1.8 * 10^{-6}$ |

**Table 4.** Tuned coefficients of WRF-Hydro/NOAH-MP for both Experiments. OVROUGHRTFAC is the overland roughness scaling factor, REFKDT is the infiltration coefficient and REFDK is the saturation of soil hydraulic conductivity

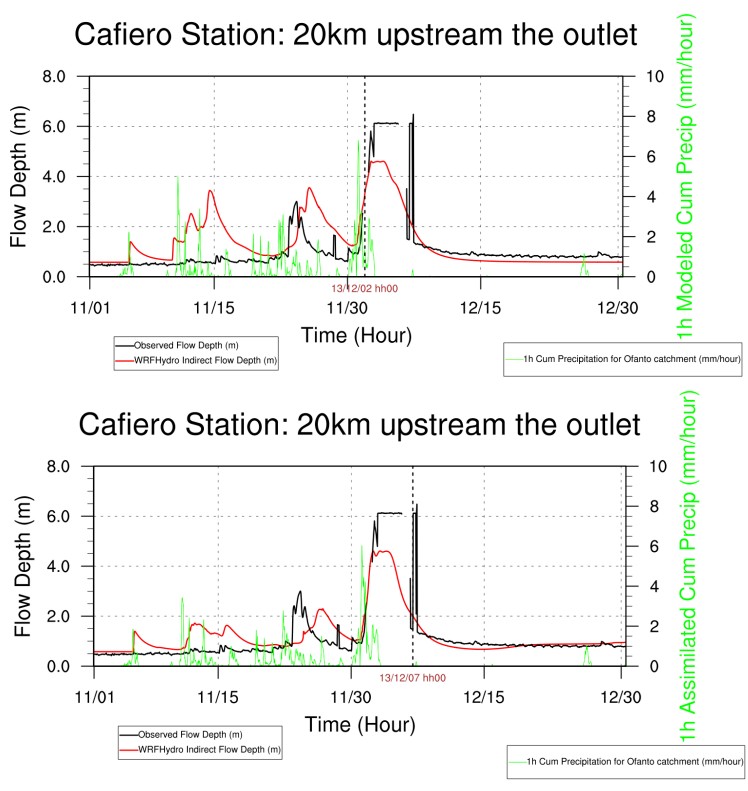

**Figure 10.** Validation of the Ofanto discharge for Experiment 2 at Cafiero Station. Top panel: modelled precipitation. Bottom panel: assimilated precipitation. The gaps in the black line are due to the river flood which damaged the gauge station

| Tunable Coefficient | Sub-basin 1 | | Sub-basins 2-3-4 | |
|---|---|---|---|---|
| | Experiment 1 | Experiment 2 | Experiment 1 | Experiment 2 |
| $Z_{ini}(mm)$ | 0.004 | 0.04 | 0.004 | 0.0004 |
| $Z_{max}(mm)$ | 1.0 | 2.0 | 2.5 | 4.0 |
| $\alpha$ | 2.9 | 2.9 | 1.9 | 1.9 |
| C $(m^3 s^{-1})$ | 0.3 | 0.02 | 0.04 | 0.003 |

**Table 5.** Tuned coefficients of WRF-Hydro/aquifer law for Experiment 1 and 2. $Z_{ini}$ is the initial value of the aquifer water depth, $Z_{max}$ is the maximum value of the aquifer water depth, $\alpha$ is the exponential law coefficient, $C$ is the volume capacity of the aquifer. Different optimal values are set depending on sub-basin and season

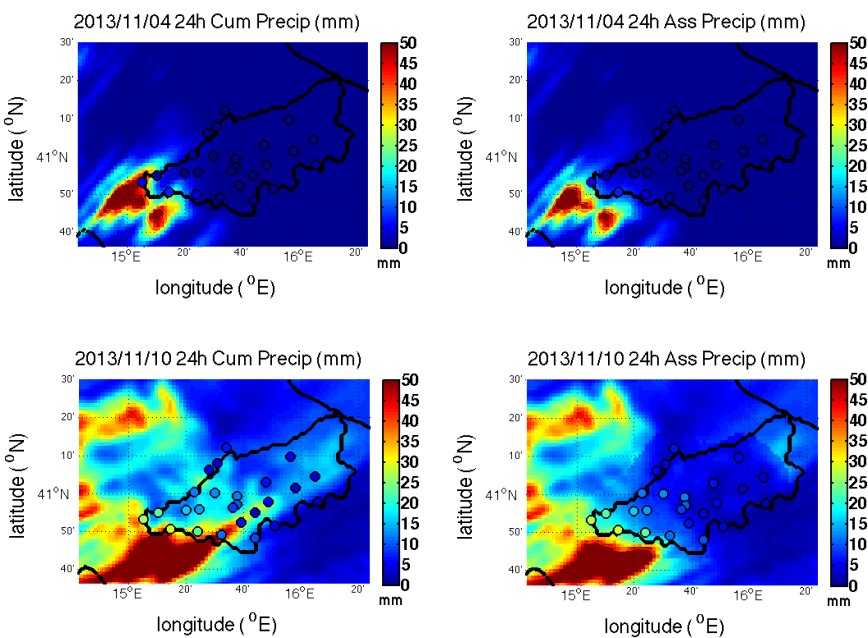

**Figure 11.** Maps of 24h cumulated precipitations (in mm/day, colours) on November, $4^{th}$ 2011 (Top panels) and November, $10^{th}$ 2013 (Bottom panels): shaded maps of modelled (left panel), and assimilated (right panel) precipitation with overlapping observed spots on the Ofanto basin

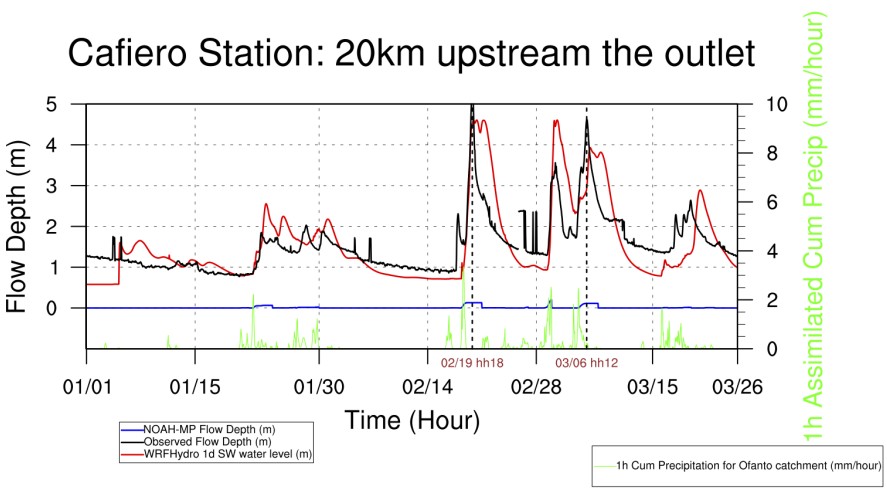

**Figure 12.** Comparison of the Ofanto discharge for Experiment 1 at Cafiero Station as provided by the best WRF-Hydro set-up and by NOAH-MP

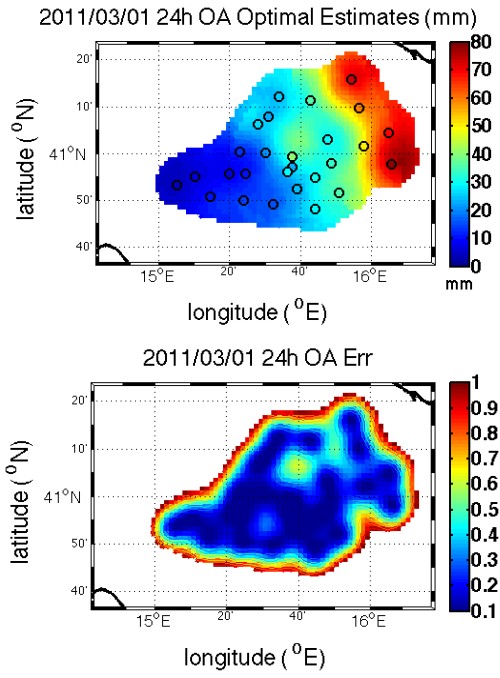

**Figure B 1.** Mapping of daily precipitation (mm/day) on 2011/03/01 as carried out by Objective Analysis. Upper panel: OA optimal estimate with overlapping observations; bottom panel: OA error

| Statistical index | modelled Precipitation | | Assimilated Precipitation | |
|---|---|---|---|---|
| of Ofanto hydrograph | Experiment 1 | Experiment 2 | Experiment 1 | Experiment 2 |
| $CV(RMSE)$ | 0.79 | 0.88 | 0.63 | 0.83 |
| $CORR$ | 0.62 | 0.72 | 0.77 | 0.86 |

**Table 6.** Statistical indices for validation of river streamflow by comparison with Cafiero-gauge station