# Peer review of "A meteo-hydrological modelling system for the reconstruction of river runoff: the case of the Ofanto river catchment"

_Natural Hazards and Earth System Sciences, 2017_

## Referee Comment (RC1) · Anonymous Referee #1 · 8 May 2017

Title: A meteo-hydrological modelling system for the reconstruction of river runoff: the case of the Ofanto river catchment Authors: Verri et al.

Recommendation: Major revisions

The paper describes the implementation, tuning and application of WRF-HYDRO to a selected watershed in southern Italy. The paper describes in detail the problems found in the implementation of the system and is interesting, especially for researchers facing with similar tasks. Anyway, I think that some points should be clarified to make the paper more mature for publication.

MAJOR POINTS: - P(page)5 L(line)7: "nested in two-way mode": in my experience

a two-way coupling is not the best way to deal with precipitation, since it improves the coarse grid results but makes worse the output in the inner grid, which is your target (you did not show the simulated precipitation in the outer domain, but I expect that it is very similar to that in the inner domain, isn't it?): did you play with these options? - P5L23: ". . . convection is assumed to have been solved explicitly, was found to perform better in the inner domain. . .": since the tuning is an important part of your study, please could you provide some additional information? In which way does the run without parameterization in the inner grid perform better? Did you try also the case with parameterization active in none (or in both) of the grids? (since you are in the grey zone for convection, it is difficult to anticipate which of these implementations would give better outputs); - P6L12: from what you write later (P8L28), I understand that an optimal range for precipitation simulation is 36-72 hours; however, from Fig. 4, it appears that the WRF runs start every 3 days, making the model skill dependent on the initial time of the simulation (i.e., a simulation starting the same day as the heavy rain will reproduce the event worse than a run starting 36 hours earlier); on the other hand, you show in Table 2 that Experiment 2 starts on the same day as the heavy rain event 2. . .: I am quite confused; - P7L30-. . .: I think the meteorological description would greatly benefit from adding mean sea level pressure contour lines in the right side of Figs. 6 and 7; also, temperature at 850 hPa is more relevant than at 2 m from a meteorological perspective; - P12L11: I do not see much change by comparing Fig.9 with Fig. 12: can you quantify the improvement?

MINOR POINTS: - P2 L20, P6L24:Âă"where the power spectrum of the turbulence reaches its peak and thus the convective motions and precipitation are only partially resolved": the fact that convection is not properly resolved is not only a consequence of turbulence, but mainly depends on the fact that the grid spacing is not sufficient to explicitly resolve the individual convective cells/systems; - P3L14:Âă" . . . characterise southern Italy . . ." - P3L29: what do you mean with "local"? Is it a single point climatology or a basin-average? - P4L14: ". . .Âăis frequently subject to lee cyclogenesis. . .": are you sure? If yes, you need to add a reference showing this point from a climatological perspective; - P4L20: a more appropriate reference for the case of November 2004 is Mastrangelo et al. (2011): Mechanisms for convection development in a long-lasting heavy precipitation event over southeastern Italy, Atmospheric Research, 100, 586-602, 2011; - P4L30: "The WRF and WRF-Hydro systems are coupled 1-way"; - P5L20 and elsewhere: YSU, not YUS; - P5L23: microphysics not mycrophysics; - P5L24: the proper reference is: Thompson et al., 2008. Explicit forecasts of winter precipitation using an improved bulk microphysics scheme. Part II: Implementation of a new snow parameterization. Mon. Weather Rev. 136: 5095–5115. The reference you put refers to an older version of the scheme. - P6L20: "uncertainties are large in mesoscale models due to unresolved meso-scale processes": although they may contribute, this is not the only reason for possible model failures; - P6L21: "grid spacing" is more appropriate than "horizontal resolution"; - P7L12: Is the OA+LS method based on 30 minute raingauge data or 24 hour cumulated rainfall? - P7L29: "...trough ... which is due to a cold front...": is it the cold front responsible for the trough or the opposite? I suggest to use "associated" instead of "due"; - P8L2: again: is the cyclone triggered by the winds or the opposite? - P8L5: "mesoscale convective systems...": I do not see mesoscale convective systems: do you mean cyclones? - P9L19: WRF-ASS: this is not really assimilation, but the result of a post-processing technique; - P11L22: are you comparing the result of your post-processing technique with the results of a simulation starting from a 3DVAR analysis? In that case, the comparison is not fair; - P11L27: "... observed water level peak ..."; - P12L20: are the flash floods really frequent in the area? Can you quantify their frequency? - P13L10: "...Âăan operational meteo-hydrological forecasting system ...": how do you think this technique can be used operationally? If you adjust the precipitation field at the initial time, you should adjust also the dynamic and thermodynamic fields to be compatible with this ...

---

## Referee Comment (RC2) · Anonymous Referee #2 · 12 May 2017

I regret to inform you that my personal opinion is that the paper needs a major revision in order to clarify the valuable work developed by its authors. To begin with, I really found confusing the description of the two objectives mentioned: the reconstruction of rainfall and runoff series. I would respectfully recommend writing two different papers in order to clarify different methodologies and perspectives unless you use runoff modelling results as criteria for rainfall estimation, which is a point that I couldn't clarify after reading your paper. I would highlight some other problems concerning the content of the paper reviewed: 1. Considering the methodology. Don't you find quite dangerous the use of a huge amount of parameters to simulate meteorological, hydrological and hydraulic processes?. What controls the over parameterization effects

on your model/system? 2. You have mentioned nothing about PET estimation, which may be an important process considering the duration of events. Have you discard its influence and why?. And considering the altitudes shown in the MDT figure and the mountainous characteristics of your catchment, I've also missed some comments about snow and melting processes. 3. Are you using a continuous model or an event based one. It seems that the actual objective is not to represent a long time series, but a collation of convective events producing extremes. 4. Would you say that long time response of aquifers is well represented in your model?. What are the physiographic characteristics of your basin in order to guess how is the aquifer response?. You also mentioned that the coefficients for the infiltration and saturated hydraulic conductivity are seasonally dependent. Don't you think that this is a problem of a lack of how your model works with water availability dependence of both infiltration and groundwater propagation laws or even more of the model capabilities but not of the parameters?. I mean, is there any structural problem in your model?. It seems that you've taken a practical approach, but not a science based one. 5. What is the main criteria to calibrate and validate rainfall and runoff?. It seems that highest extremes had been used a reference of quality. If so, why not other kind of values considering that your objective is to represent long time series? 6. Furthermore, I would say that there are many more inconsistencies that make difficult the reading of the paper. Some others may be the following ones: a. Would you say that a reconstruction from winter 2011 to autumn 2013 is a long time series? b. How do you define your catchment?. Is it a small river catchment or a medium sized one?. 7. I would recommend reviewing the conclusions section too. What I found there is a review of some topics previously described as well as some evidence previously known as the key role played by aquifer discharge to affect the baseflow. Finally, I would recommend rewriting the whole paper (or papers) considering the interesting work developed by authors and the interesting topics they have assessed based on a global modelling approach.

---

## Referee Comment (RC3) · Anonymous Referee #3 · 13 Jun 2017

General comments

The paper deal with a hydro-meteorological modelling system in order to simulate the river runoff in the Ofanto basin in the Puglia region, Southern Italy during two episodes: one between January and March 2011 and the second between November and December 2013. I found the paper interesting to be publish in the NHESS journal. The use of WRF-Hydro is wide spreading in the scientific community and I appreciate also the effort to describe in the appendix A and B mathematical formulas of algorithm and processes. However, a major review has to be done; in particular, the authors have to better clarify some parts that are not clear and are missing in the text (see below).

[Figure]

Specific comments

The paper is readable and understandable, but I suggest to take care about punctuation marks and changing of the paragraph, since some sentences are not linked between each other, see for instance P2 L25-28, P5, L1-3, P7 L 18-21. Furthermore, some parts in the text are not clear, for instance: P4, L25: Have you ever carried out a calibration and validation with this hydro-meteorological chain in previous years? Or, as it seems, you calibrate and validate only and during these two events? P8, L23-24: Why did you not maintain equal the lead time of forecast? As also reported in table 2, where it is written that for the event 1 the simulation starts 2 days before the main peak and for the event 2 on the same day? How do you conclude that "the WRF needs to be re-initialized approximately 1.5 days earlies"? Is there anything that, for the sake of brevity, it is not reported in the text? P9, L17-24: Here, you compare the results of your experiments with values of other researches carried out in different areas, basins, etc. I did not understand this comparison: have you tried the 3D-Var assimilation on the Ofanto river basin? P10, L24: In which period did you carry out the calibration? I would clarify better these parts in the text. Take care that there are lots of parameters and variable you introduced in your analysis: maybe it is better to focus on a few of them? Finally, a general check to the figures and tables is strictly recommended.

Technical Corrections

P1, L11: Add a comma after "however" and please the same in rest of the text P1, L20: remove "in" P2, L1: Please, choose to write Apulia or Puglia in the whole text P2, L4: replace "with" with "between" P2, L7: I suggest: "...validation procedures, but they need..." P2, L9: In addition, ... P2, 10: ...catchments, but... P2, L13: The term embed is it appropriate? I suggest "take into account" P2, L16: I suggest to replace "end result" with "final result" P2, L19: ...and, thus, ... P2, L22: Finally, ... P3, L9: Remove "thus" P3, L14: I suggest: ...river runoff, and the evaluation... P3, L15: I suggest: predictions P3, L16-19: I would not begin a new paragraph P3, L22: I suggest: The Ofanto river basin P3, L22: What do you want to mean with "relocatable"

P3, L25: dry season, but may... P3, L27: Please, add a space between numbers and units. Check it in the rest part of the paper. P3, L28: I would write 720 mm without year, since you wrote it is an annual mean rainfall P3, L32: I suggest: "...a small village, located at 715 m above the sea level." P4, L6: I suggest "four" in letters and not in number and in rest of the paper as well. P4, L8: In particular, the Calitri gauge... P4, L8: Replace "and" with "which" P5, L2: I would not begin a new paragraph P5, L12: after "thus", "overall", "in addition", "however", please add a comma in the text P5, L31: replace "in" with "on" P6, L18: I suggest: "...precipitation is crucial for the reconstruction..." P8, L10: I suggest "these" instead of "both" P8, L16: initialization time in small letters P9, L9: please add here a reference after Yucel and Senatore P9, L17: The acronym for the WRF-ASS, it is not introduced P10, L17. I suggest: "...0 and 1.0, where values equal to 1.0 mean that..." P12; 32: I suggest: The study also highlighted... P13, L15: Please change with "hydro-meteorological variables" P15, L6: I suggest: "...coefficients: the first... P16, L22: Please, add a space after "section". P16, L28: area not ar,ea P16, L29: remove "is" P24, Figure: the font of letters is too small. Please, increase it. The legend of the top panel goes from 0 to 4000 m a.s.l., but it seems that the highest altitude is much less: please review it. Then, is t worth to show the right panel? I cannot appreciate colours in the figure. P25, Figure 3: The hydro-meteorological modelling chain. P26, Figure 5: The coloured spots are the all available rain gauges previously shown in figure 2? Can you add the basin contour line? P27, Figure 6: I would use "dam" as unit of the geopotential height instead of "m/10" and °C instead of Cdeg. The same for Fig. 7. P28, Figure 8: replace "or" with "and" P29, Figure 9: Validation of the Ofanto discharge... P30, Figure 10: I would repeat in this figure as well the problem of missing data. The legend font is too small, also in figure 12 and 13.

---

## Author Comment (AC1) · 26 Jul 2017

Title: A meteo-hydrological modelling system for the reconstruction of river runoff: the case of the Ofanto river catchment Authors: Verri et al. Recommendation: Major revisions The paper describes the implementation, tuning and application of WRF-HYDRO to a selected watershed in southern Italy. The paper describes in detail the problems found in the implementation of the system and is interesting, especially for researchers facing with similar tasks. Anyway, I think that some points should be clarified to make the paper more mature for publication.

[Figure]

MAJOR POINTS: - P(page)5 L(line)7-P5L7: "nested in two-way mode": in my experience a two-way coupling is not the best way to deal with precipitation, since it improves the coarse grid results but makes worse the output in the inner grid, which is your target (you did not show the simulated precipitation in the outer domain, but I expect that it is very similar to that in the inner domain, isn't it?): did you play with these options? Authors: The experimental design is hereafter clarified and we provide references to several studies, which use the two-way coupling for the reconstruction of local rainfall events in the same region. The two domains set-up (Figure 1) is conceived to capture the genesis and the development of the mesoscale cyclonic patterns responsible for the heavy rain events in the coarse domain, moreover the finer grid mesh of the inner domain enables to reconstruct the local convection including the orography effects in the region of interest, i.e. the South-Eastern Italy. Overall we found that the two-way coupling mode improves the precipitation reconstruction at local scales. Several studies recommend the two-way coupling for reconstructing heavy rainfall on local scales: Miglietta et al. (2008), Moscatello et al. (2008), Federico et al. (2008), Laviola et al. (2011), Mastrangelo et al. (2011) among the others. Moreover all the studies cited above focus on rainfall events occurred in our region, i.e. the South-Eastern Italy. We provided more details in section 3 (page 5) that now reads as follows: "The two domains set-up (Figure 1) aims to capture the genesis and the development of the mesoscale cyclonic patterns responsible for the heavy rain events in the coarse domain, moreover the finer grid mesh of the inner domain enables to reconstruct the local convection including the orographic effects in the region of interest, i.e. the South-Eastern Italy. We tested different extensions and grid spacing of the coarse domain and we compared the 2-domains approach with the 1-domain only set-up. We found that a two domain, two-way coupling set up improves the reconstruction of precipitation at local scales (not shown)" In the revised manuscript we also stressed the previous studies that we considered as a benchmark for setting up our experimental design, including the two-way nesting method: "Overall our experimental design is based on the past studies of WRF for local rainfall events in the same region that stressed the two-way nesting: Miglietta

et al. (2008), Moscatello et al. (2008), Federico et al. (2008), Laviola et al. (2011), Mastrangelo et al. (2011) among the others."

- P5L23: ": : : convection is assumed to have been solved explicitly, was found to perform better in the inner domain: : :": since the tuning is an important part of your study, please could you provide some additional information? In which way does the run without parameterization in the inner grid perform better? Did you try also the case with parameterization active in none (or in both) of the grids? (since you are in the grey zone for convection, it is difficult to anticipate which of these implementations would give better outputs); Authors: In our study we tested the model sensitivity to the convection parameterization for the inner domain, looking at the numerical results in terms of the near surface atmospheric fields including the precipitation one. The validation of the precipitation field shows that the explicit convection performs better that the Kain-Fritsch parameterization scheme (Kain, 2004). This was an expected result and thus we didn't provide more details in the text. Probably the reviewer's concern about the use of the explicit convection in our inner domain (with 2km as horizontal spacing) is related to the fact that this grid spacing is only in 'convection permitting' scale range (i.e. horizontal grid spacing less that 4ăĂĽkm, as defined in the review paper by Prein et al., 2015). Several studies on severe convective weather forecasts have already documented that a grid spacing of few kilometers is sufficiently fine to ensure a successful reconstruction of convection, its mesoscale organization, and associated precipitation with no active convective parameterization scheme: Done et al. (2004), Weisman et al. (2008), Kain et al. (2008), Schwartz et al. (2009) & (2010), among the others. Focusing on our target region, similar studies based on WRF code and pointing to the reconstruction of local rainfall events (e.g. Miglietta et al., 2008; Moscatello et al., 2008; Federico et al., 2008; Mastrangelo et al 2011; Laviola et al., 2011) have already proved the benefits of working with the explicit convection in the "convection permitting" scale range. The studies cited above have been added the revised manuscript. Further details are provided in section 3.1 and they read as follows: "Our sensitivity tests shows that in the inner domain the explicit convection

works better than the convection parameterizationeven if the grid spacing is only in the 'convection permitting' scale range (Prein et al., 2015). This is documented by previous studies on severe convective weather forecasts: Done et al. (2004), Weisman et al. (2008), Kain et al. (2008), Schwartz et al. (2009) & (2010), among the others."

- P6L12: from what you write later (P8L28), I understand that an optimal range for precipitation simulation is 36-72 hours; however, from Fig. 4, it appears that the WRF runs start every 3 days, making the model skill dependent on the initial time of the simulation (i.e., a simulation starting the same day as the heavy rain will reproduce the event worse than a run starting 36 hours earlier); on the other hand, you show in Table 2 that Experiment 2 starts on the same day as the heavy rain event 2: : :: I am quite confused; Authors: We thank the reviewer for pointing out this weakness in the text, as it led to a misunderstanding that we corrected in the revised manuscript. The concatenation procedure we adopted is the one described by the reviewer: a chain of 72h runs with the reinitialization option. However we should better clarify what we found out about the optimal model spin-up time. We performed two simulation experiments, each of them done for 2 different seasons (winter 2011 and autumn 2013). They were done simply concatenating 72 hours hindcasts, re-initialized every 3 days. The first experiment contained the heavy rainfall Event 1 that was found to occur 48h later than the start time of the hindcast. The second experiment contained the heavy rainfall Event 2 that was found to occur at the the start time of one of the re-initialization hindcasts.. We performed extra WRF 72h runs to test the sensitivity of Event 1 and 2 to the initialization time and we found out that the optimal spin-up time for capturing the peak events is 1.5 days. The results of the different initialization and spin up times for Event 1 are shown in Figure 8 and commented in the text. For this reason we mentioned as one of our future plans the development of a robust WRF ensemble, which consists of overlapping chains of 72h simulations with a delayed start-time (See Conclusion section). To avoid any misunderstanding we re-wrote two sentences of section 4.2.1 and they read as follows: -Sentence at page 8 line 21 has been modified as follows: "In addition to Experiment 1 and Experiment 2 we performed extra WRF

72h runs focusing on specific events to test the sensitivity of the simulated precipitation in relation to the initialization time: the panels of Figure 8 highlight the differences between the 24h cumulated precipitation on February 18th 2011 started 14 hours and 38 hours before the rain peak of Event 1 ." -Sentence at page 8 line 26 has been modified as follows: "We conclude that our WRF model would need to be re-initialized approximately 1.5 days earlier than the start of the heavy rain events to increase skill in the predicition of precipitation. . For this reason as a future step we plan to develop a robust WRF ensemble, which consists of overlapping chains of 72h simulations with a delayed start-time" By the way we believe the underestimation of the river runoff peak triggered by Event 2 (Figure 10) is partially due to the Event 2 onset overlapping the start time of WRF 72h simulation, we added this comment in section 4.3.2: "It should be also noted that the Event 2 onset overlaps the start time of WRF 72h simulation (Table 2) and this probably affects the underestimation of the runoff peak starting on December 2nd 2013".

- P7L30-: : :: I think the meteorological description would greatly benefit from adding mean sea level pressure contour lines in the right side of Figs. 6 and 7; also, temperature at 850 hPa is more relevant than at 2 m from a meteorological perspective; Authors: We modified both Fig. 6 and Fig. 7 making them more informative as suggested by the reviewer. We added mean sea level pressure contours on the left panel and we replaced 2m Temperature with 850hPa Temperature on the right panel. We agree the pictures provide now a more comprehensive description of the Events from a meteorological perspective. Overall we slightly modified the description of the pictures as follows: "Figure 6 and Figure 7 provide the mesoscale maps of the two severe weather events occurred on March 1st 2011 (Event 1) and December 1st 2013 (Event 2). The 500hPa geopotential maps highlight how the upper level features affect the lower level cyclogenesis. WRF maps for Event 1 show a strong trough of low pressure at 500hPa centered over the Western Mediterranean Sea (left panel in Figure 6), which is due to a cold front (not shown) progressing eastward. At lower levels a strong synoptic wind, coming from the southeast and blowing over the warm Ionian Sea reaches

the Italian Peninsula (right panel in Figure 6. The left panel of Figure 7 shows the 500hPa geopotential maps for Event 2: a weak trough covers the Western Mediterranean Sea in the upper troposphere, with a small but deep cyclonic core south of Sicily. This corresponds to a strong cyclonic circulation at a lower level (right panel Figure 7) with a mslp gradient reaching 16 hPa in the cyclone eye. This cyclone is situated almost directly beneath the cutoff low in the 500hPa height field and corresponds to a southerly wind carrying warm-moist air reaching the Southern Italy and a colder wind developing downslope of the Balkans" The new figures 6 and 7 are reported below for convenience.

Figure 6. Mesoscale maps during the weather storm on 1 March 2011 (Event 1). Left panel: WRF (domain1) Geopotential height (in dam,colours) at 500hPa and mean sea level pressure (in hPa, white lines). Right panel: WRF (domain1) 850hPa Temperature (in C, colors) and 10m wind (in m/s, black arrows)

Figure 7. Mesoscale maps during the weather storm on 1 December 2013 (Event 2). Left panel: WRF (domain1) Geopotential height (in dam,colours) at 500hPa and mean sea level pressure (in hPa, white lines). Right panel: WRF (domain1) 850hPa Temperature (in C, colors) and 10m wind (in m/s, black arrows)

- P12L11: I do not see much change by comparing Fig.9 with Fig. 12: can you quantify the improvement? Authors: We calibrated the tunable coefficients of the aquifer sub-model and the final configuration is the one ensuring the best reconstruction of the river baseflow, which is associated with the low flow depth values between events in the hydrograph. In order to make a quantitative comparison between the bottom panel of Fig.9 and Fig.12 more clear, we overlapped the two hydrographs, with and without the aquifer parameterization, in the same picture (i.e. Fig.9). A better reconstruction of the minimum values of the flow depth can be detected in January 5th to 20th and on February 5th to 20th when the aquifer is switched on. In the revised manuscript we added: "On the other hand, the acquifer parametrizations do not impact the quality of the reconstruction because of the small Ofanto catchment acquifer capacity, as shown

in Fig.9 (i.e. CV(RMSE) index reduces of only 2% when the aquifer is switched on and the correlation is almost the same)". We maintained this analysis in section 4.3.2 but we removed it from the Conclusions. New Figure 9 is attached below for convenience

Figure 9. Validation of the Ofanto discharge for Experiment 1 at Cafiero Station. Top panel: modelled precipitation. Bottom panel: assimilated precipitation. The blue time-series refers to the additional experiment performed with the aquifer switched off

MINOR POINTS: - P2 L20, P6L24: "where the power spectrum of the turbulence reaches its peak and thus the convective motions and precipitation are only partially resolved": the fact that convection is not properly resolved is not only a consequence of turbulence, but mainly depends on the fact that the grid spacing is not sufficient to explicitly resolve the individual convective cells/systems; Authors: We agree, the sentence is misleading thus we tried to better clarify our point: "The grid spacing of mesoscale meteorological models does not allow to fully resolve the scales of the single convective cells/systems (Moeng et al., 2007; Shin et al., 2013)"

- P3L14:ÂËŸ a" : : : characterize southern Italy : : :" Authors: Thanks for the correction

- P3L29: what do you mean with "local"? Is it a single point climatology or a basin-average? Authors: We mean basin-average. Thanks for the remark

- P4L14: ": : is frequently subject to lee cyclogenesis: : :": are you sure? If yes, you need to add a reference showing this point from a climatological perspective; Authors: We agree and rewrote the sentence above citing the studies which investigated events occurred in the last two decades : "Concerning the meteorological modelling, the case study is located in the Southern Italy, where several heavy rainfall and flash flood events have occurred in the last decades triggered by lee cyclogenesis and convective instability (Federico et al., 2008 & 2009; Moscatello et al., 2008; Miglietta et al., 2008; Mastrangelo et al., 2011)."

- P4L20: a more appropriate reference for the case of November 2004 is Mastrangelo

et al. (2011): Mechanisms for convection development in a long-lasting heavy precipitation event over southeastern Italy, Atmospheric Research, 100, 586-602, 2011; Authors: We included this reference, thanks.

- P4L30: "The WRF and WRF-Hydro systems are coupled 1-way"; Authors: Thanks for the correction

- P5L20 and elsewhere: YSU, not YUS; Authors: Thanks for the correction

- P5L23: microphysics not mycrophysics; Authors: Thanks for the correction

- P5L24: the proper reference is: Thompson et al., 2008. Explicit forecasts of winter precipitation using an improved bulk microphysics scheme. Part II: Implementation of a new snow parameterization. Mon. Weather Rev. 136: 5095–5115. The reference you put refers to an older version of the scheme. Authors: We corrected it, thanks

- P6L20: "uncertainties are large in mesoscale models due to unresolved meso-scale processes": although they may contribute, this is not the only reason for possible model failures; Authors: Thanks for the remark. We modified this sentence to make it more general: "The simulation of the localisation, amount and timing of precipitation is crucial for the reconstruction of a river runoff time series but uncertainties are large in mesoscale models, particularly due to unresolved meso$\beta$ and meso$\gamma$ scale processes".

- P6L21: "grid spacing" is more appropriate than "horizontal resolution"; Authors: Thanks we corrected it

- P7L12: Is the OA+LS method based on 30 minute raingauge data or 24 hour cumulated rainfall? Authors: As we already explained in the section 3.2 and in the Appendix B, the OA+LS method is applied on hourly basis: the observed precipitation is recorded every 30 minutes but we used the precipitation cumulated over 1 hour.

- P7L29: ": : :trough : : : which is due to a cold front: : :": is it the cold front responsible for the trough or the opposite? I suggest to use "associated" instead of "due"; Authors: The synoptic analysis few days before the Event 1 and Event 2 indicates

the intensification of the tough at the 500hPa level is accompaniend at the surface by the strengthen of a warm-moist wind coming from the South and a cold wind on the lee-side of the Balkans,which encircle the low level cyclonic core.. We modified our misleading sentences, see major point P7L30 for the details.

- P8L2: again: is the cyclone triggered by the winds or the opposite? Authors: See the above point -P7L29.

- P8L5: "mesoscale convective systems: : :": I do not see mesoscale convective systems: do you mean cyclones? Authors: We replaced "mesoscale convective systems" with "cyclones".

- P9L19: WRF-ASS: this is not really assimilation, but the result of a post-processing technique; Authors: Thanks for the remark, we actually refer to "correction procedure" through the text. We removed "WRF-ASS precipitation" with "corrected WRF precipitation" to avoid misunderstanding.

- P11L22: are you comparing the result of your post-processing technique with the results of a simulation starting from a 3DVAR analysis? In that case, the comparison is not fair; Authors: Actually we performed a "post-processing correction method" based on the Objective Analysis plus the Least Squares Method while Yucel (2014) considers a 3D-Var assimilated field. The comparison is conceived to stress the level of accuracy of our corrected precipitation with respect to previous studies even based on more advanced correction tools as the 3D-Var.

- P11L27: ": : : observed water level peak : : :"; Authors: Corrected, thanks

- P12L20: are the flash floods really frequent in the area? Can you quantify their frequency? Authors: Thanks for the remark. We modified our statement as already explained at point -P4L14

- P13L10: ": : an operational meteo-hydrological forecasting system : : :": how do you think this technique can be used operationally? If you adjust the precipitation field

at the initial time, you should adjust also the dynamic and thermodynamic fields to be compatible with this : : : Authors: This is intended as a future step so not enough detailed in the presented study. We will keep in mind your remark.

Please also note the supplement to this comment:
https://www.nat-hazards-earth-syst-sci-discuss.net/nhess-2017-102/nhess-2017-102-AC1-supplement.pdf

[Figure]

**Fig. 1.** Figure6 Caption in the text

[Figure]

[Figure]

[Figure]

**Fig. 2.** Figure7 Caption in the text

[Figure]

**Fig. 3.** Figure9 Caption in the text

---

## Author Comment (AC3) · 26 Jul 2017

General comments The paper deal with a hydro-meteorological modelling system in order to simulate the river runoff in the Ofanto basin in the Puglia region, Southern Italy during two episodes: one between January and March 2011 and the second between November and December 2013. I found the paper interesting to be published in the NHESS journal. The use of WRF-Hydro is wide spreading in the scientific community and I appreciate also the effort to describe in the appendix A and B mathematical formulas of algorithm and processes. However, a major review has to be done; in

particular, the authors have to better clarify some parts that are not clear and are missing in the text (see below).

Specific comments The paper is readable and understandable, but I suggest to take care about punctuation marks and changing of the paragraph, since some sentences are not linked between each other, see for instance P2 L25-28, P5, L1-3, P7 L 18-21. Authors: Thanks for the remarks.

Furthermore, some parts in the text are not clear, for instance: P4, L25: Have you ever carried out a calibration and validation with this hydro-meteorological chain in previous years? Or, as it seems, you calibrate and validate only and during these two events? Authors: Preliminary sensitivity tests including CalVal activities (not shown) have been performed over several time ranges, from weekly to seasonal, for setting up the final configuration of the WRF model (Table 1). The coupled WRF-Hydro system instead, has been calibrated and validated only over the Experiment 1 and 2 periods as shown in this paper. The CalVal with observational data has been performed on the whole two periods and not only during the extreme events.

P8, L23-24: Why did you not maintain equal the lead-time of forecast? As also reported in table 2, where it is written that for the event 1 the simulation starts 2 days before the main peak and for the event 2 on the same day? How do you conclude that "the WRF needs to be re-initialized approximately 1.5 days earlier"? Is there anything that, for the sake of brevity, it is not reported in the text? Authors: We appreciated the reviewer's comment and we realized that more details are required in order to make our concatenation strategy clear. Giorgia per favore guarda il mio commento prima, a mio parere devi dire se hai sostituito nella serie temporale il re-run per gli eventi 1 e 2. We performed two simulation experiments, each of them done for 2 different seasons (winter 2011 and autumn 2013). They were done simply concatenating 72 hours hindcasts, re-initialized every 3 days. The first experiment contained the heavy rainfall Event 1 that was found to occur 48h later than the start time of the hindcast. The second experiment contained the heavy rainfall Event 2 that was found to occur at the

the start time of one of the re-initialization hindcasts.. We performed extra WRF 72h runs to test the sensitivity of Event 1 and 2 to the initialization time and we found out that the optimal spin-up for capturing the peak events is 1.5 days. These results are shown in Figure 8. For this reason we mentioned as one of our future plans the development of a robust WRF ensemble, which consists of overlapping chains of 72h simulations with a delayed start-time (See Conclusion section). To avoid any misunderstanding we re-wrote two sentences of section 4.2.1 and they read as follows: -Sentence at page 8 line 21 has been modified as follows: "In addition to Experiment 1 and Experiment 2 we performed extra WRF 72h runs focusing on specific events to test the sensitivity of the simulated precipitation in relation to the initialization time: the panels of Figure 8 highlight the differences between the 24h cumulated precipitation on February 18th 2011 started 14 hours and 38 hours before the rain peak of Event 1 ." -Sentence at page 8 line 26 has been modified as follows: "We conclude that our WRF model would need to be re-initialized approximately 1.5 days earlier than the start of the heavy rain events to increase skill in the prediction of precipitation. . For this reason as a future step we plan to develop a robust WRF ensemble, which consists of overlapping chains of 72h simulations with a delayed start-time" By the way we believe the underestimation of the river runoff peak triggered by Event 2 (Figure 10) is partially due to the Event 2 onset overlapping the start time of WRF 72h simulation, we added this comment in section 4.3.2: "It should be also noted that the Event 2 onset overlaps the start time of WRF 72h simulation (Table 2) and this probably affects the underestimation of the runoff peak starting on December 2nd 2013".

P9, L17-24: Here, you compare the results of your experiments with values of other researches carried out in different areas, basins, etc. I did not understand this comparison: have you tried the 3D-Var assimilation on the Ofanto river basin? Authors: We performed a "post-processing correction method" based on the Objective Analysis plus the Least Squares Method while Yucel (2014) considers a 3D-Var assimilated field. The comparison is conceived to stress the level of accuracy of our corrected precipitation with respect to previous studies even based on more advanced correction tools as

the 3D-Var.

P10, L24: In which period did you carry out the calibration? I would clarify better these parts in the text. Take care that there are lots of parameters and variable you introduced in your analysis: maybe it is better to focus on a few of them? Authors: The calibration procedure has been performed for the whole period of Experiment 1, January-March 2011, and Experiment 2, November-December 2013.. The strategy for calibrating such a high number of parameters consists of the two steps briefly described now in section 4.3.1. We have now explained in more detail the preliminary step of the calibration procedure (based on PEST software, see section 4.3.1) which enabled us to reduce the original set of 25 to 7 tunable parameters. The coefficients showing a high correlation (i.e. |corr|>0.9) or the ones that preserved almost the original values after the PEST run, have been excluded from the second step, i.e. the manual calibration of the reduced set of parameters. Few details on the preliminary calibration have been added in section 4.3.1 and we modified the text as follows: "As a first step we adopted an automated calibration procedure, based on the PEST software (Doherty, 2002). This procedure minimizes an objective function, given by the sum of the mean squared differences between the modelled and observed river streamflow, using the Gauss-Marquardt-Levenberg non-linear least squares method. Several tests were carried out and we identified the most relevant parameters to be calibrated for our case study. The coefficients with a high correlation (i.e. |corr|>0.9) or the ones that preserved almost the original values after the PEST tests have been excluded. Thus we reduced the original set of 25 tunable parameters to 7 that are, found to play a key role in the Ofanto basin. They are: the surface roughness scaling factor which controls the hydrograph shape and the timing of the peaks; the infiltration coefficient, the saturated hydraulic conductivity and the aquifer coefficients which control the total water volume". Finally we would like to underline that the WRF-Hydro system involves several tunable coefficients but ensures a good compromise between the number of parameterizations involved and the number of described physical processes, differently from simplified rainfall-runoff models as HEC-HSM and TOPKAPI among the others.

Finally, a general check to the figures and tables is strictly recommended. Authors: Figures 2, 5 and 11 have been modified following the reviewer's comments in the Technical Corrections.

Technical Corrections Authors: We thank the reviewer for the technical corrections. We modified the manuscript accordingly. Some clarifications are between the lines

P1, L11: Add a comma after "however" and please the same in rest of the text P1, L20: remove "in" P2, L1: Please, choose to write Apulia or Puglia in the whole text P2, L4: replace "with" with "between" P2, L7: I suggest: " ::: validation procedures, but they need ::: " P2, L9: In addition, ::: P2, 10: ::: catchments, but :: P2, L13: The term embed is it appropriate? I suggest "take into account" P2, L16: I suggest to replace "end result" with "final result" P2, L19: ::: and, thus, :: P2, L22: Finally, :: P3, L9: Remove "thus" P3, L14: I suggest: :: river runoff, and the evaluation ::: P3, L15: I suggest: predictions P3, L16-19: I would not begin a new paragraph P3, L22: I suggest: The Ofanto river basin P3, L22: What do you want to mean with "relocatable" Authors: We mean that the final configuration of our meteo-hydrological modeling chain may be easily applied to investigate rainfall and runoff events in other study areas with similar physiographic characteristics: "This is intended to be a relocatable case study as the final configuration of our meteo-hydrological modeling chain may be easily applied to investigate rainfall and runoff events in other study areas with similar physiographic characteristics".

P3, L25: dry season, but may ::: P3, L27: Please, add a space between numbers and units. Check it in the rest part of the paper. P3, L28: I would write 720 mm without year, since you wrote it is an annual mean rainfall P3, L32: I suggest: " ::: a small village, located at 715 m above the sea level." P4, L6: I suggest "four" in letters and not in number and in rest of the paper as well. P4, L8: In particular, the Calitri gauge ::: P4, L8: Replace "and" with "which" P5, L2: I would not begin a new paragraph P5, L12: after "thus", "overall", "in addition", "however", please add a comma in the text P5, L31: replace "in" with "on" P6, L18: I suggest: " ::: precipitation is crucial for the

reconstruction ::: " P8, L10: I suggest "these" instead of "both" P8, L16: initialization time in small letters P9, L9: please add here a reference after Yucel and Senatore P9, L17: The acronym for the WRF-ASS, it is not introduced P10, L17. I suggest: " ::: 0 and 1.0, where values equal to 1.0 mean that ::: " P12; 32: I suggest: The study also highlighted:: P13, L15: Please change with "hydro-meteorological variables" P15, L6: I suggest: " ::: coefficients: the first ::: P16, L22: Please, add a space after "section". P16, L28: area not ar,ea P16, L29: remove "is" P24, Figure: the font of letters is too small. Please, increase it. The legend of the top panel goes from 0 to 4000 m a.s.l., but it seems that the highest altitude is much less: please review it. Then, is t worth to show the right panel? I cannot appreciate colours in the figure. Authors: We modified the upper value of the colorbar for the top panel of Figure 2, thanks for your suggestion. The new picture is attached below for convenience.

Figure 2. The Ofanto River Catchment. Top panel: Topography height (units of m) and location of 27 rain-gauge stations

We maintained the bottom right panel of Fig.2, because it gives an idea of the GIS procedure we adopted for drawing the river network and the hierarchy of tributaries with the colorbar showing the number of draining cells and thus the flow directions.

P25, Figure 3: The hydro meteorological modelling chain. Authors: The coauthor David Gochis recommends the expression "meteo-hydrological modeling chain" instead.

P26, Figure 5: The coloured spots are the all available rain gauges previously shown in figure 2? Can you add the basin contour line? Authors: Figure 5 as well as Figure 11 have been modified following your suggestion. The new pictures are attached below for convenience.

Figure 5. Maps of 24h cumulated precipitations (in mm/day, colours) during the peak events on March, 1st 2011 (Top panels) and December,1st 2013 (Bottom panels): shaded maps of modelled (left panel), and assimilated (right panel) precipitation with overlapped observed spots over the Ofanto basin

Figure 11. Maps of 24h cumulated precipitations (in mm/day, colours) on November, 4th 2011 (Top panels) and November, 10th 2013 (Bottom panels): shaded maps of modelled (left panel), and assimilated (right panel) precipitation with overlapping observed spots on the Ofanto basin

P27, Figure 6: I would use "dam" as unit of the geopotential height instead of "m/10" and C instead of Cdeg. The same for Fig. 7. P28, Figure 8: replace "or" with "and" P29, Figure 9: Validation of the Ofanto discharge ::: P30, Figure 10: I would repeat in this figure as well the problem of missing data. The legend font is too small, also in figure 12 and 13

Please also note the supplement to this comment:
https://www.nat-hazards-earth-syst-sci-discuss.net/nhess-2017-102/nhess-2017-102-AC3-supplement.pdf

[Figure]

**Fig. 1.** Figure2 Caption in the text

[Figure]

**Fig. 2.** Figure5 Caption in the text

[Figure]

**Fig. 3.** Figure11 Caption in the text

---

## Author Comment (AC2)

I regret to inform you that my personal opinion is that the paper needs a major revision in order to clarify the valuable work developed by its authors. To begin with, I really found confusing the description of the two objectives mentioned: the reconstruction of rainfall and runoff series. I would respectfully recommend writing two different papers in order to clarify different methodologies and perspectives unless you use runoff modelling results as criteria for rainfall estimation, which is a point that I couldn't clarify after reading your paper. I would highlight some other problems concerning the content of the paper reviewed:

**Authors:**

A reliable description of the Ofanto river hydrograph goes through a proper description of the meteorological and soil processes, with the precipitation field playing the most relevant role. Details and references (Pappenberger et al., 2005; Zappa et al., 2010) about this approach are common in the literature and they are referenced in the Introduction.

Overall the causality dependence between a proper reconstruction of the meteorological and the hydrological events is the reason why we decided to consider both topics in the same paper. We have inserted a phrase in the Introduction that explicitly states:

"*In this paper we describe both the precipitation reconstruction and the hydrograph results since the former drive most of the quality of the hydrology of a river basin*".

1. Considering the methodology. Don't you find quite dangerous the use of a huge amount of parameters to simulate meteorological, hydrological and hydraulic processes?. What controls the over parameterization effects on your model/system?

**Authors:**

WRF-Hydro is one of the most advanced hydrological/hydraulics system currently available in the literature. We agree that this system includes a large number of tunable coefficients; on the other hand it ensures a good compromise between the number of described physical processes and the number of involved parameterizations.

We have now explained in more detail the preliminary step of the calibration procedure (based on PEST software, see section 4.3.1) which enabled us to reduce the original set of 25 to 7 tunable parameters. The coefficients showing a high correlation (i.e. |corr|>0.9) or the ones that preserved almost the original values after the PEST run, have been excluded from the second step, i.e. the manual calibration of the reduced set of parameters.

Details on the preliminary step of our calibration procedure have been added in section 4.3.1 and we modified the previous text as follows:

"*As a first step we adopted an automated calibration procedure, based on the PEST software (Doherty, 2002). This procedure minimizes an objective function, given by the sum of the mean squared differences between the modelled and observed river streamflow, using the Gauss-Marquardt-Levenberg non-linear least squares method. Several tests were carried out and we identified the most relevant parameters to be calibrated in our specific case study. The coefficients with a high correlation (i.e. |corr|>0.9) or the ones that preserved almost the original values after the PEST tests have been excluded. Thus we reduced the original set of 25 tunable parameters to 7 that are found to play a key role in the Ofanto basin. They are: the surface roughness scaling factor which controls the hydrograph shape and the timing of the peaks; the infiltration coefficient, the saturated hydraulic conductivity and the aquifer coefficients which control the total water volume.*"

Finally we added the following sentence in the Conclusions:

"*More research is still required on the groundwater modeling as it greatly impacts the overland waterflow and the river runoff but also the evapotranspiration and consequently the precipitation. We plan to evaluate different parameterizations of the aquifer recharge/discharge. Overall a reduction of the parameterizations involved in the WRF-Hydro system could be desirable*"

2. You have mentioned nothing about PET estimation, which may be an important process considering the duration of events. Have you discard its influence and why?. And considering the altitudes shown in the MDT figure and the mountainous characteristics of your catchment, I've also missed some comments about snow and melting processes.

**Authors:**

The NOAH-MP v2.7.1 model uses a modified Penman's relationship for the potential evapotranspiration, PET (Mahrt and Ek, 1984). The Penman's formula for the PET is widely used and we didn't perform any check on that.

As already detailed in section 3.1, we rather focused on the land use and topography datasets as we found that the low atmosphere and land surface fields including the evapotranspiration are strongly dependent on them, thus we replaced the default USGS data with the higher resolution and more recent data released by the European Environmental Agency.

Snow modeling is active in the NOAH-MP model starting from the ECMWF accumulated snow depth data. However, over the whole watershed, the highest point is at 1100m a.s.l.thus the snowfall and the melting processes might be of minor importance in the Ofanto catchment. We have now modified Fig.2 to make evident the height of the terrain in the catchment. The new picture is attached below for convenience.

[Figure]

Figure 2. The Ofanto River Catchment. Top panel: Topography height (units of m) and location of 27 rain-gauge stations

We added more details in the manuscript :

*"The snow modeling is also active in NOAH-MP model: a multilayer snow pack, the snow albedo, the melting/refreezing capability are solved by NOAH-MP. Moreover the evaporation component coming from the snow sublimation is added and the evaporation component coming from the canopy water is split into the rainfall and the snowfall terms. The ECMWF analyses used for computing the initial and boundary conditions provide also the accumulated snow depth at the groundlevel. For our case studies, the snowfall and the melting processes do not seem to play a crucial role".*

3. Are you using a continuous model or an event based one. It seems that the actual objective is not to represent a long time series, but a collation of convective events producing extremes.

**Authors:**

Our experimental design is bases a concatenation of 3-days hindcasting experiments. Details on the concatenation strategy and simulation period are provided below.

Literature proved that seasonal and sub-seasonal meteorological reconstructions benefit from a frequent reinitialization, i.e. few days (Qian et al., 2003; Koster et al., 2010; Lucas-Picher et al., 2013). On the other hand the hydrological-hydraulics reconstruction need a longer spin up period to distribute the overland and subsurface waterflow and to allow the river network to reach a steady state starting from dry at the initial state (Senatore et al., 2015), thus the WRF hydrological components are started from dry conditions and not restarted for the whole first month of concateneted meteorological forcing.

To make our strategy clear, we added few details at the end of section 3.1:

*"The hydraulics component of WRF-Hydro system is initialized with the NOAH MP overland and subsurface water flows that are dry at the initial time. Thus a spin-up period is required to laterally route the groundwater of the basin and to allow the river network to reach a steady state. Senatore et al. (2015) considered monthly spin-up for evaluating the WRF-Hydro results and we decided to follow the same strategy."*
We have also  added Koster et al. (2010) to the references about the WRF re-initialization strategy.

4. Would you say that long time response of aquifers is well represented in your model?. What are the physiographic characteristics of your basin in order to guess how is the aquifer response?. You also mentioned that the coefficients for the infiltration and saturated hydraulic conductivity are seasonally dependent. Don't you think that this is a problem of a lack of how your model works with water availability dependence of both infiltration and groundwater propagation laws or even more of the model capabilities but not of the parameters?. I mean, is there any structural problem in your model?. It seems that you've taken a practical approach, but not a science based one.
**Authors:**
In the presented study we focused on evaluating the model capability to reconstruct the precipitation and the river runoff. The aquifer representation is important in so far as it feeds the river baseflow and the numerical outcomes prove that the results are not extremely dependent on the acquifer specific parametrizations.
In general we followed a practical approach. The physiographic characteristics of the basin support our practical approach as we expect the soil types impact the capability of the aquifer to discharge to stored water (as already highlighted in section 4.3.1). Overall we agree that the empirical formula for the aquifer recharge/discharge and the high number of tunable coefficients affecting the groundwater are far from being satisfactory as we stressed at the end of section 4.3.1:
*"This study found that the soil infiltration and the aquifer water storage parametrizations should be seasonally dependent. This means that the present parameterizations of these processes are not capable to capture the complexity of the groundwater physical processes"*.
Additionally we have metioned this problem in the Conclusions:
*"More research is required to establish a better groundwater modeling that at the moment considers seasonally dependent, ad hoc values of the soil infiltration and the aquifer water storage. We plan to evaluate different parameterizations of the aquifer recharge/discharge. Overall a reduction of the parameterizations involved in the WRF-Hydro system could be desirable"*

5. What is the main criteria to calibrate and validate rainfall and runoff? It seems that highest extremes had been used a reference of quality. If so, why not other kind of values considering that your objective is to represent long time series?
**Authors:**
For both precipitation and runoff, all the available observations covering the whole time windows of the experiments, including both extreme events and dry periods, have been used to calibrate the tunable coefficients and to validate the modeling results. We were interested both in the hydrograph baseflow and peak events which clearly are the most important for societal impact.  Following the comments of the first reviewer we have modified Fig. 9 adding the experiments with and without the acquifer switched on since we were trying to simulate also the baseflow.

6. Furthermore, I would say that there are many more inconsistencies that make difficult the reading of the paper. Some others may be the following ones: a. Would you say that a reconstruction from winter 2011 to autumn 2013 is a long time series? b. How do you define your catchment?. Is it a small river catchment or a medium sized one?.
**Authors:**
We believe we have already answered these questions since:

a)  we performed two seasonal experiments over winter 2011 and autumn 2013 and not a unique reconstruction from winter 2011 to autumn 2013. Thus our reconstructions produce relatively short-term timeseries. We hope our answer to point 3 has clarified the strategy adopted for the experimental design.

b) As we already stated in section 2, the Ofanto catchment is a medium size catchment (i.e. between 1000 and 10000 $km^2$) because it is about 2790$km^2$.

7. I would recommend reviewing the conclusion section too. What I found there is a review of some topics previously described as well as some evidence previously known as the key role played by aquifer discharge to affect the baseflow. Finally, I would recommend rewriting the whole paper (or papers) considering the interesting work developed by authors and the interesting topics they have assessed based on a global modelling approach.

**Authors:**

We reviewed the conclusions following the reviewer's suggestions and we entitled this section "*Summary, conclusions and future plans*" since it also includes an overview of the main modeling findings of the presented study. We list no more the aquifer influence on the river baseflow among our findings, as this is an expected outcome and the small storage capacity of the Ofanto aquifer makes the statistics on the aquifer influence unimportant.

The following sentences have been added aiming at making the purpose of this study more clear:

"*Overall we highlighted the 2-way feedback existing between a proper reconstruction of the meteorological events and the hydrological ones. A reliable description of the river hydrograph goes through a proper description of the meteorological and soil processes, with the precipitation field playing the most relevant role. At the same time the validation of the river hydrograph works as a effective post-processing tool to calibrate the water infiltration through the soil column and the aquifer recharge/discharge as well as to correct the modeled precipitation with the OA+LS method*".

"*More research is required to establish a better groundwater modeling that at the moment considers seasonally dependent, ad hoc values of the soil infiltration and the aquifer water storage. We plan to evaluate different parameterizations of the aquifer recharge/discharge. Overall a reduction of the parameterizations involved in the WRF-Hydro system could be desirable*".